# Unsupervised Zero-Shot Reinforcement Learning via Dual-Value Forward-Backward Representation

**Jingbo Sun[1,2,3], Songjun Tu[1,2,3], Qichao Zhang[1,3],[*] Xin Liu[1,3], Haoran Li[1,3], Yaran Chen[4] Ke Chen[2], Dongbin Zhao[1,2,3]**
[1]Institute of Automation, Chinese Academy of Sciences, [2]Pengcheng Laboratory
[3]University of Chinese Academy of Sciences, [4]Xi'an Jiaotong-Liverpool University
{sunjingbo2022,tusongjun2023,zhangqichao2014}@ia.ac.cn
{liuxin2021,lihaoran2015,dongbin.zhao}@ia.ac.cn
chenk02@pcl.ac.cn, yaran.chen@xjtlu.edu.cn

## Abstract

Online unsupervised reinforcement learning (URL) can discover diverse skills via reward-free pre-training and exhibits impressive downstream task adaptation abilities through further fine-tuning. However, online URL methods face challenges in achieving zero-shot generalization, i.e., directly applying pre-trained policies to downstream tasks without additional planning or learning. In this paper, we propose a novel **D**ual-**V**alue **F**orward-**B**ackward representation (DVFB) framework with a contrastive entropy intrinsic reward to achieve both zero-shot generalization and fine-tuning adaptation in online URL. On the one hand, we demonstrate that poor exploration in forward-backward representations can lead to limited data diversity in online URL, impairing successor measures, and ultimately constraining generalization ability. To address this issue, the DVFB framework learns successor measures through a skill value function while promoting data diversity through an exploration value function, thus enabling zero-shot generalization. On the other hand, and somewhat surprisingly, by employing a straightforward dual-value fine-tuning scheme combined with a reward mapping technique, the pre-trained policy further enhances its performance through fine-tuning on downstream tasks. Through extensive experiments, DVFB demonstrates both superior zero-shot generalization (outperforming on all 12 tasks) and fine-tuning adaptation (leading on 10 out of 12 tasks) abilities, surpassing state-of-the-art (SOTA) URL methods. Our code is available at `https://github.com/bofusun/DVFB`.

## 1 Introduction

In recent years, deep reinforcement learning (DRL) (Sutton & Barto, 1999) has achieved remarkable success in various fields, including game AI (Tang et al., 2023), autonomous driving (Wu et al., 2022), robot control (Li et al., 2024). However, most DRL approaches rely on predefined task-specific rewards for training, which limits the generalizability of the learned policies to new tasks. Given the varied demands of real-world applications, it is crucial to develop agents capable of addressing multiple tasks directly, as in zero-shot reinforcement learning (zero-shot RL) (Touati et al., 2023), or through task adaptation. This need has spurred growing interest in training task-agnostic policies that can generalize across tasks, a paradigm known as unsupervised reinforcement learning (URL). In this paper, we aim to achieve downstream task generalization in online URL with both zero-shot capability and the potential for further improvement through fine-tuning.

Previous research (Eysenbach et al., 2019; Laskin et al., 2022; Park et al., 2024) has explored unsupervised skill discovery (USD) for task generalization, focusing on developing diverse skills via reward-free learning to enable task adaptation through fine-tuning. However, USD methods aim to maximize the divergence between the state distribution of skills and the average state distribution, posing challenges for zero-shot generalization. Other studies, such as Barreto et al. (2017); Touati et al. (2023) utilize successor representation (SR) techniques (Dayan, 1993) to facilitate zero-shot

---

[*]Corresponding author.

generalization. SR methods can be categorized into successor features (SF) (Barreto et al., 2017) and forward-backward representations (FB) (Touati & Ollivier, 2021). They define skill vectors based on successor representations, enabling them to infer near-optimal skills from minimal demonstrations of downstream tasks. Unfortunately, SR methods rely on pre-collected diverse datasets (Jeen et al., 2024) and are constrained to offline settings. Our findings further reveal their poor zero-shot generalization performance in online URL. These insights trigger our further thought: Can the SR mechanism, such as FB, be extended to online URL to enable zero-shot generalization capability?

Extending the zero-shot generalization of forward-backward representations to online URL presents several challenges. First, we find that FB mechanism typically exhibit limited exploration capabilities, resulting in reduced data diversity. This limitation hinders the effectiveness of successor measures and further restricts zero-shot generalization in downstream tasks. Second, FB mechanism rely on implicit rewards that have a fixed relationship with skill during pre-training, which prevents the direct use of intrinsic rewards to enhance online exploration, as USD methods do. Therefore, the key to extending FB to online URL for zero-shot generalization is developing a mechanism that enhances policy exploration for diverse buffer data while maintaining efficient successor representation and skill learning.

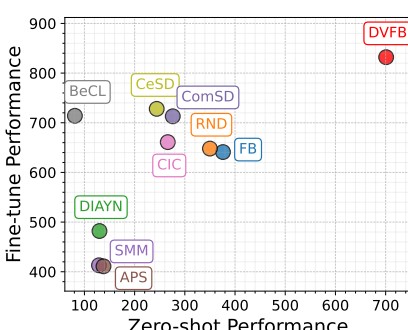

Figure 1: DVFB achieves leading zero-shot and fine-tuning performances in *Walker* and *Quadruped* Domain.

In this paper, we explore the underlying reasons for the failure of the FB mechanism in online URL. Our findings reveal that insufficient exploration in FB leads to low data diversity, which impedes the learning of successor measures and results in inaccurate value functions for downstream tasks, ultimately limiting zero-shot capabilities. To address these challenges, we propose a **D**ual-**V**alue **F**orward-**B**ackward representation (DVFB) framework, which leverages a skill value function to learn forward-backward representations while incorporating an exploration value function to encourage exploration. We introduce a novel intrinsic objective based on contrastive learning, which enhances exploration while learning distinguishable state transitions, thereby improving zero-shot generalization across various downstream tasks. Additionally, we introduce a dual-value fine-tuning scheme with a reward mapping technique that leverages both the pre-trained skill value and the downstream task value to ensure performance improvement beyond zero-shot capabilities.

In summary, our contributions are as follows:

- We investigate the reasons for FB's failure in online URL and find that its inadequate exploration leads to low data diversity. This hinders the learning of successor measures and the value network, ultimately limiting zero-shot generalization capabilities.
- We propose a novel dual-value forward-backward representation (DVFB) architecture with an intrinsic reward based on contrastive learning. The pre-trained DVFB agents can zero-shot generalize to various downstream tasks.
- We introduce a dual-value fine-tuning scheme with a reward mapping technique, enhancing performance stability during downstream task fine-tuning beyond zero-shot capabilities by leveraging the pre-trained skill value function with forward-backward representations.
- Our method is comprehensively validated across twelve challenging robot control tasks in the DeepMind Control Suite, demonstrating superior generalization performance. To the best of our knowledge, our proposed DVFB is the first method to achieve both zero-shot generalization and fine-tuning capabilities in online URL, as shown in Figure 1.

## 2 RELATED WORK

### 2.1 ZERO-SHOT TASK GENRALIZATION IN RL

Zero-shot task generalization in RL refers to an agent's ability to apply its knowledge directly to new tasks after training, without any additional training (Touati et al., 2023; Jeen et al., 2024). Methods

for addressing task generalization in RL include model-based RL, multi-task RL, unsupervised skill discovery, and successor representations. Model-based reinforcement learning (RL) (Chua et al., 2018; Hafner et al., 2021) trains a task-agnostic world model to assist in decision-making; however, it still requires a reward function to plan for new tasks. Multi-task RL (He et al., 2024; Lan et al., 2024) enables generalization across predefined related tasks but cannot be applied to unfamiliar tasks. USD (Eysenbach et al., 2019; Laskin et al., 2022) learns diverse skills in reward-free environments, but its objective focuses on maximizing the distance between the skill's state distribution and the average state marginal distribution. Consequently, USD is unable to achieve zero-shot generalization (Eysenbach et al., 2022). None of these methods can achieve zero-shot generalization.

Successor feature (SF) (Dayan, 1993) methods establish successor features using handcrafted or learned reward-related features $\phi$, which can then be zero-shot generalized to downstream tasks (Barreto et al., 2017; Zhang et al., 2017). To relax the linear assumption of $\phi$ and reward in SF, successor measures directly learn future state distributions and avoid the need for $\phi$ (Touati & Ollivier, 2021). FB methods (Touati et al., 2023) achieve zero-shot generalization by learning successor measures from pre-collected diverse data. FRE (Frans et al., 2024) focuses on learning functional representations by encoding state-reward samples with a variational auto-encoder to achieve zero-shot generalization. However, these methods require access to offline datasets for pre-training, which cannot be expected for most real problems (Jeen et al., 2024). In addition, they show high data sensitivity and limited performance on pre-collected fixed dataset (see detailed analysis in Appendix G). Extending FB to the online URL setting without the offline dataset collection process is a promising direction. Unfortunately, we find it fails to achieve zero-shot generalization in online URL. Hence, this paper proposes a DVFB framework that extends FB methods to online URL for both zero-shot generalization and efficient task adaptation to downstream tasks.

## 2.2 Unsupervised Pre-training in RL

Unsupervised pre-training methods in reinforcement learning can be divided into two categories: online and offline. This paper primarily focuses on the former. Online URL typically involves learning useful representations or skills for downstream tasks through intrinsic rewards. Based on the modeling of intrinsic rewards, URL can be further divided into unsupervised skill discovery (USD) methods(Eysenbach et al., 2019; Sharma et al., 2020; Park et al., 2022; Laskin et al., 2022; Park et al., 2024) and data coverage maximization methods (Liu & Abbeel, 2021b; Yarats et al., 2021; Lee et al., 2019). Existing USD methods leverage mutual information as an intrinsic reward to learn diverse skills that can rapidly adapt to downstream tasks(Eysenbach et al., 2019; Sharma et al., 2020; Liu et al., 2025; Laskin et al., 2022). Some studies(Eysenbach et al., 2022; Yang et al., 2024) indicate that maximizing mutual information is equivalent to maximizing the difference between the average state distribution and the most distinct skill state distribution. As a result, although these methods enhance fine-tuning by providing optimal initialization, they lack the capacity for zero-shot generalization to downstream tasks. Some studies attempt to improve state coverage with Euclidean distance constraints(Park et al., 2022), Wasserstein distance constraints(Park et al., 2024), and guided skills(Kim et al., 2024). However, they lack a clear definition of skills and their connection to downstream task rewards, which hinders zero-shot generalization.

## 3 Preliminaries

### 3.1 Reward-free markov decision processes and genralization in URL

In unsupervised RL, a reward-free Markov Decision Process (MDP) (Sutton & Barto, 1999; Touati et al., 2023) is defined as the tuple $M = \langle S, A, P, \gamma \rangle$, where $S$ is the state space, $A$ is the action space, and $P(ds'|s, a)$ is the probability measure of the next state $s' \in S$, which defines the stochastic transition from state $s$ to state $s'$ after taking an action $a$. $\gamma$ is the discount factor. Given state-action pair $(s_0, a_0) \in \mathcal{S} \times \mathcal{A}$ and a policy $\pi : \mathcal{S} \to \text{Prob}(\mathcal{A})$, we use $\Pr(\cdot \mid s_0, a_0, \pi)$ and $\mathbb{E}(\cdot \mid s_0, a_0, \pi)$ to represent the probability and expectation of the state-action sequence $(s_t, a_t)_{t \geq 0}$ obtained by starting from the initial state $(s_0, a_0)$ and using the policy $\pi$, where states are sampled as $s_t \sim P(ds_t \mid s_{t-1}, a_{t-1})$ and actions are sampled as $a_t \sim \pi(da_t \mid s_t)$. The state transition probability under the policy $\pi$ is defined as $P^\pi(ds' \mid s) = \int P(ds' \mid s, a)\pi(da \mid s)$. Given a reward function $r : \mathcal{S} \to \mathbb{R}$, the $Q$-function of a policy $\pi$ with respect to the reward $r$ is defined as $Q_r^\pi(s_0, a_0) := \sum_{t \geq 0} \gamma^t \mathbb{E}[r(s_t) \mid s_0, a_0, \pi]$.

During the pre-training phase in URL, the agent interacts with a reward-free environment and learns skills using data collected by periodically updating the skills. At the testing time, the policy aims to maximize the expected discounted return for the task, given by $\mathbb{E}_{t \geq 0}[\gamma^t r_{eval}(s_{t+1}) \mid s_0, a_0, \pi]$. For zero-shot generalization, the agent must infer skills from a small dataset $D_{\text{infer}} = \{(s_t, r(s_t))\}_{t=1}^k$, where $k < 10000$, obtained through limited interactions with the downstream task environment.

### 3.2 THE FORWARD-BACKWARD REPRESENTATIONS

Following (Touati & Ollivier, 2021), the successor measure $M^\pi(s_0, a_0, \cdot)$ represents the cumulative discounted time spent in each state $s_{t+1}$, starting from a state-action pair $(s_0, a_0)$ and following the policy $\pi$, namely:

$$M^\pi(s_0, a_0, X) := \sum_{t \geq 0} \gamma^t \Pr(s_{t+1} \in X \mid s_0, a_0, \pi), \quad \forall X \subseteq \mathcal{S} \tag{1}$$

The FB framework provides an approximately optimal policy for any reward by learning a tractable representation of the successor measure. Let $\mathcal{R}^d$ be a representation space, and let $\rho$ be an arbitrary distribution over states. The FB framework estimates the successor measure $M^{\pi_z}$ by learning a forward representation $F : \mathcal{S} \times \mathcal{A} \times \mathcal{R}^d \to \mathcal{R}^d$, a backward representation $B : \mathcal{S} \to \mathcal{R}^d$, and a set of skill policies $(\pi_z)_{z \in \mathcal{R}^d}$, such that:

$$\begin{cases} M^{\pi_z}(s_0, a_0, X) \approx \int_X F(s_0, a_0, z)^\top B(s)\rho(ds), & \forall s_0 \in \mathcal{S}, a_0 \in \mathcal{A}, X \subseteq \mathcal{S}, z \in \mathcal{R}^d, \\ \pi_z(s) \approx \arg\max_a F(s, a, z)^\top z, & \forall (s, a) \in \mathcal{S} \times \mathcal{A}, z \in \mathcal{R}^d. \end{cases} \tag{2}$$

If the approximation(2) holds, for any reward function $r$, the policy $\pi_{z_r}$ is optimal for $r$, with the optimal $Q$-function given by: $Q_r^* = F(s, a, z_r)^\top z_r$. The near-optimal skill is represented as $z_r = B(s')r$. However, FB's limited exploration capability makes it challenging to sample diverse data in online URL setting, hindering the validity of the approximation in (2) for all states. Therefore, it struggles to maintain zero-shot generalization ability in the online setting.

## 4 WHAT HINDERS THE APPLICATION OF FB TO ONLINE URL?

**Does FB's exploration affects zero-shot performance in online URL?** Unlike offline URL, where the agent learns skills from pre-collected diverse data, in online URL the agent is trained using self-generated trajectories through online exploration. Consequently, whether FB's exploration ability impacts its zero-shot performance in online URL remains a question. To explore this impact, we first evaluate FB's online exploration performance. Specifically, we compare the motion trajectories generated by FB with those produced by random network distillation (RND), a pure exploration method, within the Walker domain. The trajectories for FB are generated by randomly sampling 20 skills after 1 million steps of pre-training, with each skill repeated four times, whereas RND generates trajectories 80 times. The walking distances for the different trajectories of the RND and FB agents are presented in Figure 2(a) and (b), respectively. In the figure, lines of the same color represent trajectories corresponding to the same skill. Compared to RND, FB explores only

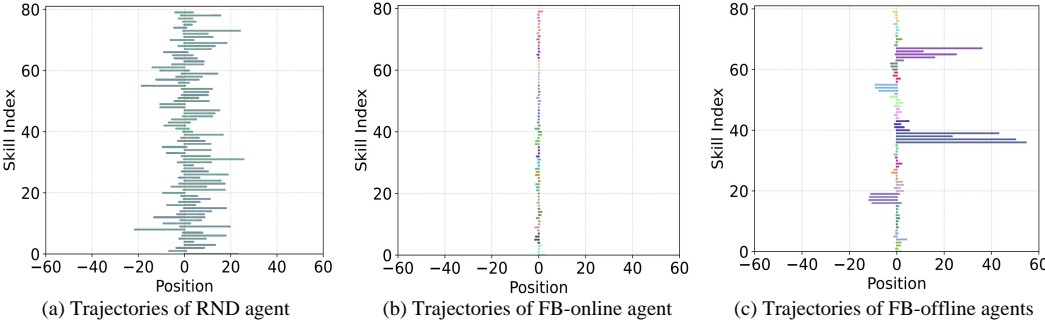

(a) Trajectories of RND agent     (b) Trajectories of FB-online agent     (c) Trajectories of FB-offline agents

Figure 2: **The trajectories of the RND, FB-online, and FB-offline agents in the *Walker walk* task.** The FB-online agent learns short-range trajectories, in contrast to the RND agent's diverse exploration and the FB-offline agent's ability to master long-range locomotion skills.

short-distance trajectories, resulting in less diverse exploration outcomes. Thus, we infer that insufficient exploration in FB leads to limited data diversity, which impacts its zero-shot generalization. To verify this conjecture, we trained the FB using the diverse dataset generated by a well-trained RND agent, as described in (Touati et al., 2023), which we refer to as FB-offline. Figure 2(c) illustrates the trajectories of different skills from FB-offline, some of which exhibit significant positional changes. Furthermore, we conduct an additional experiment on a specific downstream task: *Walker Walk*, with the training curves shown in Figure 3(a). The zero-shot generalization performance of FB-offline improves with training, while FB-online, lacking any prior data, consistently performs poorly. Therefore, we conclude that the impact of exploration on data diversity limits FB's zero-shot generalization ability.

**How do exploration and data diversity affect FB's zero-shot performance?** FB achieves zero-shot generalization by ensuring that the successor representation satisfies Equation 2 for all states. However, in the online URL, only the states that are explored meet this requirement.

If high-reward states related to downstream tasks remain unexplored, it may result in inaccurate successor measures and value estimates, ultimately affecting the capability for zero-shot generalization. To validate this hypothesis, we compare the accuracy of the value functions composed of successor measures for FB-online and FB-offline in the *Walker walk* task after pretraining. As shown at the top of Figure 3(b), we analyze the normalized values of trajectories across different ranges of episode rewards. The value function of FB-online estimates higher values for low-return trajectories, while FB-offline assigns a higher value for high-return trajectories. This suggests that FB-offline, trained on diverse data, possesses accurate successor measures and value functions for downstream tasks, whereas FB-online, trained on self-generated data, does not. Additionally, we introduce Spearman correlation (Spearman, 1987) to reflect the correlation between the prior skill value $\left(Q_M = F(s, a, z)^\top z\right)$ and the episode rewards of trajectories. Spearman correlation measures the strength and direction of a monotonic relationship between two ranked variables. A higher Spearman correlation suggests that the inferred value function of the policy aligns more closely with the rewards of the downstream tasks. As shown at the bottom of Figure 3(b), FB-offline exhibits improved Spearman correlation across all trajectories, while FB-online displays a poor correlation with the episode rewards of high-return trajectories. This finding indicates that FB-online's successor measures for downstream tasks are inaccurate. Therefore, we conclude that exploration hinders FB's zero-shot ability by impacting its successor measure of high-reward states.

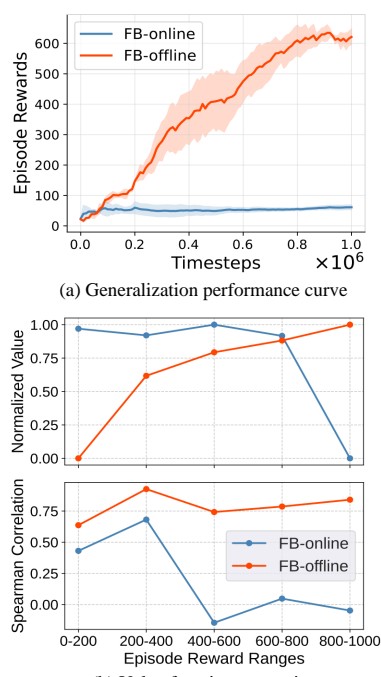

(a) Generalization performance curve

(b) Value function properties

Figure 3: Generalization curve and value function properties. (a) presents the agents' performance during pre-training, while (b) illustrates the normalized skill value and the Spearman correlation between skill value and return across different return ranges.

## 5 METHODOLOGY

### 5.1 DUAL-VALUE FORWARD-BACKWARD REPRESENTATION PRE-TRAINING FRAMEWORK

To enhance the exploration capability of forward-backward representations, a straightforward idea is to incorporate exploration-based intrinsic rewards. The forward-backward representation achieves zero-shot generalization by learning the successor measure $M^\pi$ that satisfies a Bellman-like equation: $M^\pi = P + \gamma P_\pi M^\pi$. To satisfy Bellman's equation, the value function for skill $z$ is associated with a fixed implicit reward, given by: $r_{implicit} = B^\top (E_\rho B B^\top)^{-1} z$ (Touati et al., 2023). Therefore, directly training the value function $Q_M$ using intrinsic rewards for exploration, which are in conflict with $r_{implicit}$ for FB representations, would impede the skill learning.

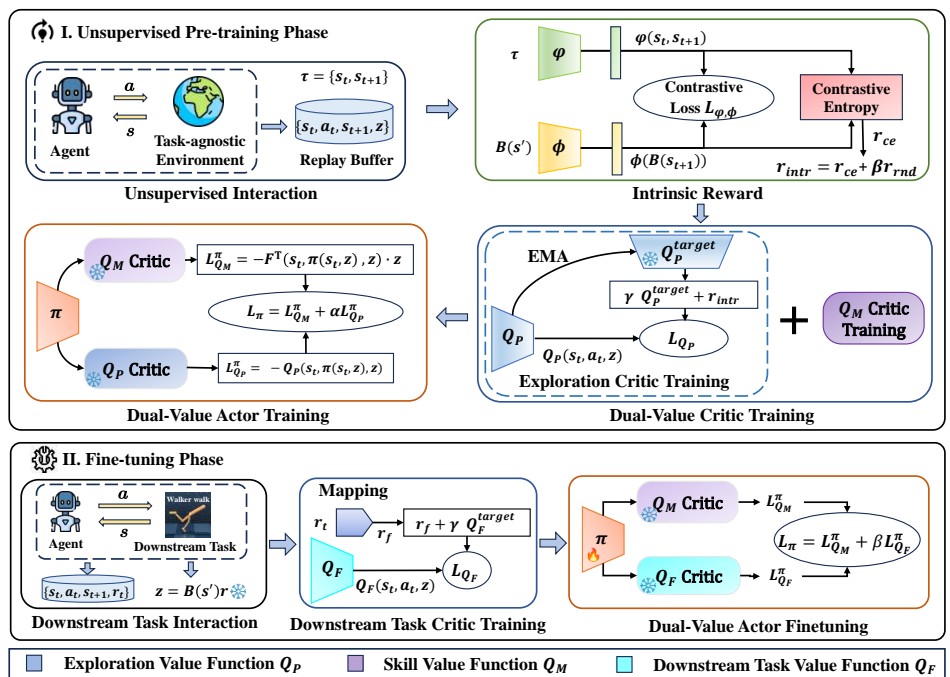

Figure 4: **The overall framework of DVFB.** During the pre-training, the exploration Critic $Q_P$ and skill Critic $Q_M$ are trained and utilized to train the actor. During the fine-tuning, skills $z$ are inferred, and the downstream task Critic $Q_F$ is trained with mapped extrinsic reward $r_f$. Finally, the policy $\pi$ is fine-tuned with the downstream task Critic $Q_F$ and the prior skill Critic $Q_M$.

**Dual-Value Pre-training Architecture.** To address this reward conflict problem, we propose a dual-value network architecture including the exploration value and skill value. The pseudocode for pre-training is presented in Appendix A. Specifically, we train an additional exploration value function $Q_P$ to promote policy exploration. The training objective for $Q_P$ is defined as:

$$L(Q_P) = \mathbb{E}_{(s_t, a_t, z) \sim \mathcal{D}}[(Q_P(s_t, a_t, z) - (r_{intr} + \gamma \overline{Q_P}(s_{t+1}, a_{t+1}, z)))^2], \quad (3)$$

where $r_{intr}$ represents the intrinsic reward that encourages exploration, $\mathcal{D}$ refers to the replay buffer, and $\overline{Q_P}$ denotes the target networks of $Q_P$. In parallel, we train the forward-backward representations to learn the skill value $Q_M = F(s_t, a_t, z)^\top z$. The training objective for forward-backward representations is given by:

$$L(F, B) = \mathbb{E}_{(s_t, a_t, s_{t+1}, z) \sim \mathcal{D}, s' \sim \mathcal{D}} \left[ \left( F(s_t, a_t, z)^\top B(s') - \gamma \overline{F}(s_{t+1}, \pi(s_{t+1}, z)^\top \overline{B}(s'))^2 \right] \right.$$
$$\left. - 2\mathbb{E}_{(s_t, a_t, s_{t+1}, z) \sim \mathcal{D}} \left[ F(s_t, a_t, z)^\top B(s_{t+1}) \right], \right. \quad (4)$$

where $s'$ represents the future state of $s_t$, $\overline{F}$ and $\overline{B}$ are the target networks for the forward and backward representations, respectively. We update the policy network by utilizing the exploration value function $Q_P$ and skill value function $Q_M$. The training objective for the policy network $\pi$ is:

$$L(\pi) = \mathbb{E}_{a_t \sim \pi_\zeta(\cdot | s_t, z)}[-F(s_t, a_t, z)^\top z - \alpha Q_P(s_t, a_t, z)], \quad (5)$$

where $\alpha$ is a coefficient. This dual-value architecture encourages the policy $\pi$ to explore diverse data in online URL guided by exploration value $Q_P$ without impairing its skill learning capabilities. Consequently, the DVFB agent learns successor measures for various states using diverse buffer data, thereby enabling zero-shot generalization for downstream tasks with different reward functions.

**Contrastive Entropy Intrinsic Reward.** Although pure exploration rewards such as RND improve policy exploration by focusing exclusively on unseen states, they overlook the discriminability of learned skills during exploration, which is helpful for FB-based skill learning. We introduce a novel contrastive entropy reward to encourage skill discrimination during exploration. First, we utilize the intrinsic reward $r_{rnd}$ from RND as a pure exploration reward:

$$r_{rnd} = ||f(s_{t+1}) - \overline{f}(s_{t+1})||^2, \quad (6)$$

where $f$ is the representation network of states, and $\overline{f}$ is the target network of $f$ with frozen parameters. Then, we train the encoders $\varphi$ and $\phi$ for state transitions and skills using contrastive learning to enhance skill discriminability. This approach improves the similarity between state transitions and their corresponding skills by updating the encoders with NCE loss (Gutmann & Hyvärinen, 2010) within a contrastive learning framework (Chen et al., 2020). Since skills derived from historical samples may not satisfy Equation 2 due to disturbances caused by exploration, our method leverages skills inferred from the next state to train a skill discriminator that aligns with the forward-backward representation. The training objective of the encoders is:

$$L(\varphi, \phi) = \frac{\varphi(\tau_i)^\top \phi(B(s_{t+1})_i)}{\|\varphi(\tau_i)\|\|\phi(B(s_{t+1})_i)\|T} - \log \frac{1}{N} \sum_{j=1}^{N} \exp\left(\frac{\varphi(\tau_j)^\top \phi(B(s_{t+1})_i)}{\|\varphi(\tau_j)\|\|\phi(B(s_{t+1})_i)\|T}\right), \quad (7)$$

where $\varphi$ and $\phi$ are the encoders for the transition $\tau$ and skill $z = B(s_{t+1})$, $T$ is the temperature parameter, $\tau$ denotes a state transition $(s_t, s_{t+1})$, $N$ is the number of contrastive pairs, and $i$ and $j$ are their indices. We leverage the trained encoders to compute similarities in contrastive entropy, enhancing skill discernibility by using particle estimates of dissimilarity entropy for each state transition as discriminative intrinsic rewards. The contrastive entropy intrinsic reward $r_{ce}$ is:

$$r_{ce} \propto \frac{1}{N_k} \sum_{z^* \in N_k} \log\left(2 - \frac{\varphi(\tau_i)^\top \phi(z^*)}{\|\varphi(\tau_i)\|\|\phi(z^*)\|T}\right), \quad (8)$$

where $z^*$ is a kNN skill, $N_k$ is the number of kNNs. The total intrinsic reward $r_{intr}$ is:

$$r_{intr} = r_{rnd} + \beta r_{ce}, \quad (9)$$

where $\beta$ is a coefficient. We combine the pure exploration reward and contrastive entropy reward as intrinsic rewards, enhancing exploration while keeping the skill learning ability of FB. Furthermore, we offer a theoretical analysis to support the zero-shot generalization of DVFB in Appendix H.

## 5.2 DUAL-VALUE FORWARD-BACKWARD REPRESENTATION FINE-TUNING SCHEME

**Dual-Value Fine-tuning Scheme.** Furthermore, we propose a dual-value fine-tuning scheme that leverages prior skill value function and zero-shot generalized policy to achieve improved performance through fine-tuning. During fine-tuning, we use environmental rewards from downstream tasks to train a task value function $Q_F$, guiding the agent to learn policies with improved task-specific performances. The training objective for the downstream task value function is as follows:

$$L(Q_F) = \mathbb{E}_{(s_t, a_t, z) \sim \mathcal{D}}[\left(Q_F(s_t, a_t, z) - (r_f + \gamma \overline{Q_F}(s_{t+1}, a_{t+1}, z))\right)^2], \quad (10)$$

where $r_f$ is the downstream task reward used to fine-tune. Since the downstream task value function is trained from scratch, its inaccuracy during the early stages of fine-tuning can disrupt the zero-shot initial policy, hindering fine-tuning efficiency. To address this issue, we incorporate prior knowledge from the pre-training phase to mitigate this impact. Specifically, We fine-tune the policy network using both the downstream task value function $Q_F$ and the prior skill value function $Q_M(s, a, z) = F(s, a, z)^\top z$. The training objective of the policy network during fine-tuning is as follows:

$$L(\pi) = \mathbb{E}_{a_t \sim \pi_\zeta(\cdot|s_t, z)}[-\eta F(s_t, a_t, z)^\top z - Q_F(s_t, a_t, z)], \quad (11)$$

where $z$ is the skill inferred from minimal demonstrations in the downstream task, $\eta$ is a coefficient. In the early fine-tuning stage, the downstream task value function, trained from scratch, holds minimal values. In contrast, the prior skill value function maintains stable values, thereby assuming a more significant role and encouraging the policy to preserve effective zero-shot performance. In the later fine-tuning phase, the downstream task value function also exhibits stable values, fostering further improvements in the agent's zero-shot performance. Thus, the dual-value fine-tuning scheme ensures stable performance improvements over the zero-shot baseline.

**Reward Mapping Technique.** After pre-training, the forward-backward representation selects different skills for zero-shot generalization across multiple downstream tasks, resulting in task-specific variations in the prior implicit rewards $r_{implicit} = B^\top (E_\rho BB^\top)^{-1} z$. The range of prior implicit rewards relative to downstream task rewards significantly impacts multi-task fine-tuning performance. Excessively high prior implicit rewards can restrict the policy's ability to surpass its zero-shot performance, whereas overly low implicit rewards may undermine the zero-shot policy, leading to

a relearning process from scratch. To mitigate the impact of the prior implicit reward range on fine-tuning performance, we propose a reward mapping technique that balances the importance of rewards by mapping downstream task rewards into the range of prior implicit rewards. During fine-tuning, we dynamically track the means of both prior implicit rewards and downstream task rewards, adjusting the downstream task rewards to align with the range of prior implicit rewards as follows:

$$r_f = r_t \frac{\mu(r_{implicit})}{\mu(r_t)}, \tag{12}$$

where $r_f$ represents the mapped downstream task reward for fine-tuning, $r_t$ refers to downstream task environmental reward, and $\mu$ denotes the dynamically computed mean of the reward. Through the reward mapping, the agent achieves stable performance improvements during fine-tuning.

## 6 EXPERIMENTS

In this section, we conduct experiments to address these questions: (1) Does DVFB exhibit zero-shot generalization in online URL? (2) Can DVFB build upon zero-shot generalization for efficient and stable fine-tuning? (3) What is the impact of various modules on generalization performance?

### 6.1 DMC CONTINUOUS CONTROL TASKS

**Environment Setup.** Following the latest advancements (Yang et al., 2023; Bai et al., 2024), we evaluate task generalization performance using 12 downstream tasks across 3 domains in URLB (Laskin et al., 2021) and DeepMind Control Suite (DMC) (Tassa et al., 2018). Details of tasks can be found in Appendix C.1.

**Baselines.** We compare DVFB with several strong baselines for unsupervised task generalization. In zero-shot generalization, we evaluate DVFB against (i) successor feature methods, including SF-CL (Balestriero & LeCun, 2022), SF-LRA-SR (Touati et al., 2023), SF-LRA-P (Ren et al., 2023), SF-Lap (Wu et al., 2018), and FB (Touati et al., 2023), and (ii) skill discovery methods, including CIC (Laskin et al., 2022), BeCL (Yang et al., 2023), ComSD (Liu et al., 2025), and CeSD (Bai et al., 2024). For fine-tuning, we compare DVFB with (i) pure exploration methods, including RND (Burda et al., 2019) and Disagreement (Pathak et al., 2019), and (ii) skill discovery methods, including SMM (Lee et al., 2019), DIAYN (Eysenbach et al., 2019), APS (Liu & Abbeel, 2021a), CIC, BeCL, ComSD, and CeSD. details and clear classification are given in Appendix C.2.

**Does DVFB exhibit zero-shot generalization in online URL?** We evaluate the zero-shot performance of DVFB across 12 tasks in the DMC. During pre-training, each method is trained for 2 million steps without rewards. Subsequently, each agent interactes with environment for $1e^4$ steps to infer the skills most suitable for downstream tasks. As shown in Table 1, our method exhibits significantly improved zero-shot performance across all tasks and domains. In the challenging *Hopper* domain, DVFB exhibits enhanced zero-shot generalization performance, while all other baseline methods struggle to acquire valuable skills. Notably, the zero-shot performance of DVFB remains competitive, even when compared to the fine-tuning performance of USD methods in Table 2. The

Table 1: DMC Zero-shot Performance

| Domain | Task | CL | Lap | LRA-P | LRA-SR | FB | CIC | BeCL | ComSD | CeSD | **Ours** |
|---|---|---|---|---|---|---|---|---|---|---|---|
| Walker | Stand | $170 \pm 82$ | $497 \pm 84$ | $425 \pm 83$ | $212 \pm 127$ | $303 \pm 55$ | $357 \pm 26$ | $152 \pm 34$ | $297 \pm 9$ | $217 \pm 71$ | $\mathbf{905 \pm 27}$ |
| | Walk | $45 \pm 19$ | $118 \pm 25$ | $82 \pm 20$ | $53 \pm 27$ | $48 \pm 14$ | $179 \pm 8$ | $29 \pm 2$ | $124 \pm 13$ | $64 \pm 34$ | $\mathbf{900 \pm 53}$ |
| | Flip | $53 \pm 18$ | $112 \pm 35$ | $75 \pm 23$ | $56 \pm 25$ | $49 \pm 8$ | $213 \pm 14$ | $27 \pm 6$ | $162 \pm 7$ | $105 \pm 65$ | $\mathbf{515 \pm 67}$ |
| | Run | $34 \pm 17$ | $88 \pm 14$ | $80 \pm 11$ | $38 \pm 21$ | $55 \pm 14$ | $78 \pm 3$ | $28 \pm 1$ | $60 \pm 3$ | $65 \pm 13$ | $\mathbf{423 \pm 53}$ |
| | Average | 76 | 204 | 166 | 90 | 114 | 207 | 59 | 161 | 113 | **686** |
| Quadruped | Stand | $539 \pm 304$ | $438 \pm 64$ | $352 \pm 9$ | $335 \pm 206$ | $897 \pm 76$ | $469 \pm 87$ | $154 \pm 85$ | $526 \pm 60$ | $516 \pm 11$ | $\mathbf{953 \pm 15}$ |
| | Walk | $272 \pm 153$ | $184 \pm 47$ | $185 \pm 63$ | $166 \pm 131$ | $453 \pm 42$ | $241 \pm 36$ | $67 \pm 44$ | $312 \pm 99$ | $268 \pm 34$ | $\mathbf{624 \pm 53}$ |
| | Jump | $409 \pm 218$ | $289 \pm 58$ | $251 \pm 80$ | $261 \pm 162$ | $716 \pm 58$ | $359 \pm 51$ | $115 \pm 63$ | $438 \pm 56$ | $452 \pm 16$ | $\mathbf{816 \pm 19}$ |
| | Run | $288 \pm 154$ | $217 \pm 38$ | $179 \pm 61$ | $170 \pm 102$ | $450 \pm 43$ | $231 \pm 33$ | $77 \pm 42$ | $283 \pm 50$ | $264 \pm 9$ | $\mathbf{467 \pm 18}$ |
| | Average | 377 | 282 | 242 | 233 | 638 | 325 | 103 | 390 | 375 | **715** |
| Hopper | Hop | $0 \pm 0$ | $0 \pm 0$ | $0 \pm 0$ | $0 \pm 0$ | $2 \pm 2$ | $4 \pm 1$ | $0 \pm 0$ | $1 \pm 0$ | $2 \pm 2$ | $\mathbf{73 \pm 18}$ |
| | Flip | $0 \pm 0$ | $1 \pm 1$ | $0 \pm 0$ | $0 \pm 0$ | $1 \pm 1$ | $7 \pm 2$ | $1 \pm 0$ | $1 \pm 0$ | $1 \pm 1$ | $\mathbf{110 \pm 18}$ |
| | Hop backward | $0 \pm 0$ | $1 \pm 1$ | $0 \pm 0$ | $2 \pm 5$ | $2 \pm 1$ | $16 \pm 3$ | $1 \pm 0$ | $5 \pm 1$ | $3 \pm 2$ | $\mathbf{125 \pm 24}$ |
| | Flip backward | $0 \pm 0$ | $1 \pm 1$ | $0 \pm 0$ | $0 \pm 0$ | $8 \pm 8$ | $15 \pm 3$ | $1 \pm 0$ | $4 \pm 1$ | $4 \pm 4$ | $\mathbf{96 \pm 8}$ |
| | Average | 0 | 1 | 0 | 1 | 3 | 11 | 1 | 3 | 3 | **101** |

Table 2: DMC Fine-tune Performance

| Domain | Task | Disagreement | RND | SMM | DIAYN | APS | CIC | BeCL | ComSD | CeSD | Ours |
|---|---|---|---|---|---|---|---|---|---|---|---|
| Walker | Stand | $753 \pm 87$ | $901 \pm 19$ | $886 \pm 18$ | $789 \pm 48$ | $702 \pm 67$ | $959 \pm 2$ | $952 \pm 2$ | $962 \pm 9$ | $960 \pm 3$ | $\mathbf{972 \pm 5}$ |
| | Walk | $516 \pm 142$ | $783 \pm 35$ | $792 \pm 42$ | $450 \pm 37$ | $547 \pm 38$ | $903 \pm 21$ | $883 \pm 34$ | $918 \pm 32$ | $834 \pm 34$ | $\mathbf{961 \pm 5}$ |
| | Flip | $335 \pm 22$ | $506 \pm 29$ | $500 \pm 28$ | $361 \pm 10$ | $448 \pm 36$ | $641 \pm 26$ | $611 \pm 18$ | $630 \pm 41$ | $541 \pm 17$ | $\mathbf{927 \pm 12}$ |
| | Run | $213 \pm 32$ | $403 \pm 16$ | $395 \pm 18$ | $184 \pm 23$ | $176 \pm 18$ | $450 \pm 19$ | $387 \pm 22$ | $447 \pm 64$ | $337 \pm 19$ | $\mathbf{548 \pm 29}$ |
| | Average | 454 | 648 | 643 | 446 | 468 | 738 | 708 | 739 | 668 | $\mathbf{852(+15\%)}$ |
| Quadruped | Stand | $512 \pm 115$ | $839 \pm 25$ | $266 \pm 48$ | $718 \pm 81$ | $435 \pm 68$ | $700 \pm 55$ | $875 \pm 33$ | $824 \pm 86$ | $919 \pm 11$ | $\mathbf{965 \pm 7}$ |
| | Walk | $358 \pm 49$ | $517 \pm 41$ | $154 \pm 36$ | $506 \pm 55$ | $385 \pm 76$ | $621 \pm 69$ | $743 \pm 68$ | $735 \pm 140$ | $889 \pm 23$ | $\mathbf{908 \pm 21}$ |
| | Jump | $403 \pm 86$ | $626 \pm 23$ | $167 \pm 30$ | $498 \pm 45$ | $389 \pm 72$ | $565 \pm 44$ | $727 \pm 15$ | $686 \pm 66$ | $755 \pm 14$ | $\mathbf{831 \pm 20}$ |
| | Run | $346 \pm 32$ | $439 \pm 7$ | $142 \pm 28$ | $347 \pm 47$ | $201 \pm 40$ | $445 \pm 36$ | $535 \pm 13$ | $500 \pm 103$ | $\mathbf{586 \pm 25}$ | $536 \pm 27$ |
| | Average | 405 | 605 | 182 | 517 | 353 | 583 | 720 | 686 | 787 | $\mathbf{804(+2\%)}$ |
| Hopper | Hop | $\mathbf{74 \pm 19}$ | $67 \pm 12$ | $5 \pm 7$ | $3 \pm 4$ | $1 \pm 1$ | $59 \pm 60$ | $5 \pm 7$ | $40 \pm 35$ | $10 \pm 15$ | $\mathbf{74 \pm 17}$ |
| | Flip | $108 \pm 26$ | $97 \pm 38$ | $29 \pm 16$ | $7 \pm 8$ | $3 \pm 4$ | $96 \pm 64$ | $13 \pm 15$ | $61 \pm 47$ | $48 \pm 49$ | $\mathbf{116 \pm 18}$ |
| | Hop backward | $231 \pm 69$ | $239 \pm 35$ | $29 \pm 57$ | $9 \pm 28$ | $2 \pm 0$ | $172 \pm 64$ | $40 \pm 72$ | $92 \pm 105$ | $117 \pm 124$ | $\mathbf{273 \pm 33}$ |
| | Flip backward | $173 \pm 7$ | $\mathbf{203 \pm 16}$ | $19 \pm 34$ | $2 \pm 1$ | $10 \pm 23$ | $154 \pm 70$ | $22 \pm 36$ | $59 \pm 63$ | $74 \pm 71$ | $189 \pm 11$ |
| | Average | 147 | 152 | 21 | 5 | 4 | 120 | 20 | 63 | 62 | $\mathbf{163(+7\%)}$ |

experimental results demonstrate that DVFB exhibits excellent zero-shot generalization capabilities in online URL. Further comparison with offline zero-shot methods is provided in Appendix G.

**Can DVFB build upon zero-shot generalization for efficient and stable fine-tuning?** We evaluate the fine-tuning performance of DVFB across 12 tasks with the same pre-training setup as zero-shot generalization. As shown in Table 2, DVFB demonstrates superior performance across all three domains compared to current SOTA methods. Specifically, DVFB surpasses baseline methods by 15%, 2%, and 7% in the *Walker*, *Quadruped*, and *Hopper* domains, respectively. Unlike previous URL methods, DVFB possesses outstanding zero-shot performance and pre-trained prior knowledge. We further explore whether DVFB can utilize these advantages to improve performance in downstream tasks fine-tuning. We compare the performance variations of DVFB with 8 SOTA URL methods during fine-tuning on the *Walker* and *Quadruped* domains. As shown in Figure 5, DVFB exhibits outstanding initial performance across most environments compared to SOTA methods. Through fine-tuning, its performance further improves upon the initial performance with lower variance, demonstrating more stable and efficient fine-tuning compared to SOTA methods.

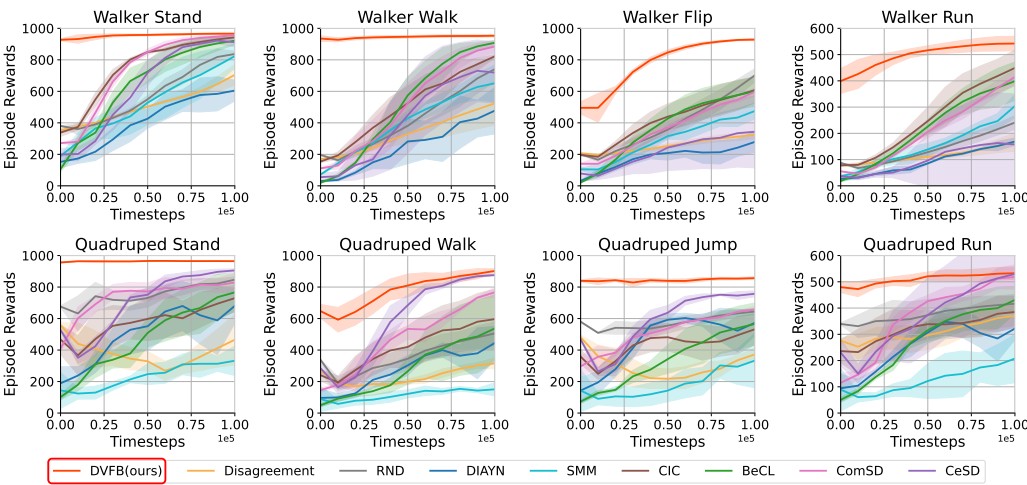

Figure 5: **Finetuning Curve on the *Walker* and *Quadruped* Domains.** DVFB begins with outstanding zero-shot performance and further enhances performance efficiently through fine-tuning.

## 6.2 WHAT IS THE IMPACT OF VARIOUS MODULES ON GENERALIZATION PERFORMANCE?

**Ablation study on the dual-value pre-training framework.** We evaluate the zero-shot generalization performance across 12 tasks using 5 seeds for pre-training and analyze the Spearman correlation and value accuracy of the skill value function to evaluate its capability in learning the successor measure. As shown in Figure 6 (a), DVFB w/o CE represents DVFB with only RND rewards as the intrinsic reward. The task names in the figure left are abbreviations of domain and task. Compared to FB's failure in most tasks, both DVFB w/o CE and DVFB achieve zero-shot generalization across

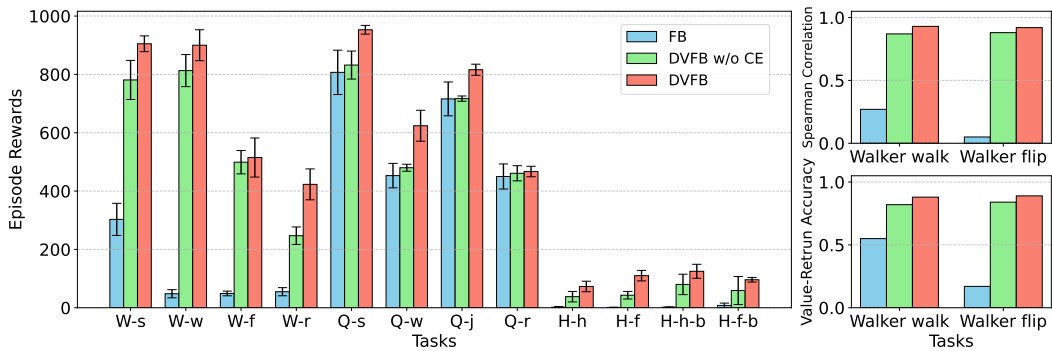

Figure 6: Results of dual-value pre-training framework ablation study.

all tasks. As illustrated in (b), the absence of the dual-value framework results in the skill value function exhibiting poor Spearman correlation and value accuracy with the downstream episode rewards. The experiments show that the proposed dual-value pre-training framework is essential for effectively learning the successor measure and facilitates zero-shot generalization. Compared to DVFB w/o CE, DVFB exhibits improved learning of successor measure and achieves higher episode rewards across all tasks. Thus, the proposed contrastive entropy intrinsic reward encourages exploration while ensuring the skill learning capability of FB, further enhancing zero-shot generalization.

**Ablation study on the dual-value fine-tuning scheme.** To evaluate the contributions of various components within the dual-value fine-tuning scheme, we compare the performance curves of DVFB using various fine-tuning methods across 4 tasks in 3 domains. As shown in Figure 7, DVFB w/o SVF employs a task value function without the prior skill value function for fine-tuning, while DVFB w/o MAP refers to DVFB without reward mapping. The figure shows that the performance of DVFB w/o SVF declines during the early stage of fine-tuning. In contrast, DVFB and DVFB w/o MAP maintain their initial performance while achieving further improvements. It shows that the dual-value fine-tuning scheme ensures stable performance improvements over the zero-shot baseline. DVFB w/o MAP improves slowly because the downstream reward values are lower than those of the implicit reward. In contrast, DVFB rapidly enhances its zero-shot performance through fine-tuning, since the reward mapping technique enables stable and rapid improvements by aligning the range of downstream and implicit rewards. More ablation results are given in Appendix F.

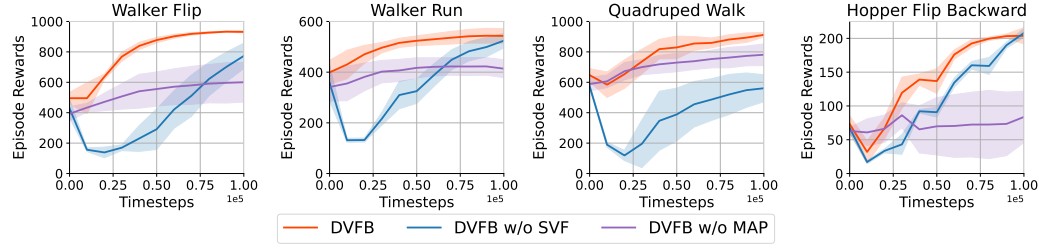

Figure 7: Results of dual-value fine-tuning scheme ablation study.

## 7 CONCLUSION

This paper introduces a general framework for achieving both zero-shot generalization and task adaptation in online URL. We first demonstrate that insufficient exploration in FB restricts the zero-shot generalization capability of the FB mechanism in online URL. To address this issue, we propose a dual-value forward-backward framework (DVFB) that encourages exploration while preserving the learning of successor measures, enabling zero-shot generalization to downstream tasks. Surprisingly, within the dual-value scheme, DVFB can further enhance performance with a reward-mapping technique by building upon its zero-shot capabilities. Notably, to the best of our knowledge, the proposed DVFB approach is the first method to simultaneously achieve zero-shot generalization and fine-tuning capabilities in online URL. The method is evaluated on 12 robot control tasks, demonstrating superior performance in both zero-shot generalization and fine-tuning compared to current SOTA methods. Although DVFB has demonstrated remarkable zero-shot generalization and fine-tuning abilities, this study mainly focuses on the URL problem within state space. Exploring the extension of the forward-backward approach to visual URL offers a promising direction.

## ACKNOWLEDGMENTS

We extend our gratitude to Fan Yang and Mingshuang Luo for their valuable and insightful discussions during the preparation of this paper. This work is supported by the National Key Research and Development Program of China under Grants 2022YFA1004000, the Beijing Natural Science Foundation under No. 4242052, the National Natural Science Foundation of China under Grants 62173325, and the CAS for Grand Challenges under Grants 104GJHZ2022013GC.

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

## A PSEUDO-CODE

We provide the complete pseudocode for DVFB, with the unsupervised pre-training phase described in Algorithm 1 and the downstream task fine-tuning phase described in Algorithm 2.

### A.1 PRE-TRAIN PSEUDO-CODE

---

**Algorithm 1** DVFB Algorithm: Unsupervised Pre-training Phase

---

1: **Inputs:** replay buffer $\mathcal{D}$, randomly initialized networks $F_\theta$, $B_\omega$, RND representation network $f_\kappa$, transition representation network $\varphi_\nu$, skill representation network $\varphi_\xi$, exploration value function $Q_{P,\sigma}$, actor network $\pi_\zeta$ , learning rate $\eta$, mini-batch size $b$, number of episodes $E$, number of gradient updates $N$, skill update period $T$, temperature $\tau$, and regularization coefficient $\lambda$.
2: **for** $m = 1$ **do**
3:     /* Collect $E$ episodes
4:     **for** $e = 1$ to $E$ **do**
5:        Sample $z \sim p(z)$
6:        Observe an initial state $s_0$
7:        **for** $t = 1$ to $T$ **do**
8:           **if** $t \mod T = 0$ **then**
9:              Sample $z \sim p(z)$
10:           **end if**
11:           Select action $a_t = \pi_\zeta(s_t, a_t, z)$
12:           Observe next state $s_{t+1}$
13:           Store transition $(s_t, a_t, s_{t+1})$ in the replay buffer $\mathcal{D}$
14:        **end for**
15:     **end for**
16:     /* Perform $N$ stochastic gradient descent updates
17:     **for** $n = 1$ to $N$ **do**
18:        Sample a mini-batch of transitions $\{(s_i, a_i, s_{i+1}, z)\}_{i \in I} \subset \mathcal{D}$ of size $|I| = b$.
19:        Sample a mini-batch of target state-action pairs $\{(s'_i, a'_i)\}_{i \in I} \subset \mathcal{D}$ of size $|I| = b$.
20:        Sample a mini-batch of $\{z_i\}_{i \in I} \sim p(z)$ of size $|I| = b$.
21:        Compute RND loss function $L(\kappa) = ||f_\kappa(s_{t+1}) - \overline{f}(s_{t+1})||_2^2$.
22:        Update $\kappa \leftarrow \kappa - \eta\nabla L(\kappa)$.
23:        Compute contrastive loss function $L(\nu, \xi)$ with equation 7.
24:        Update $\nu, \xi \leftarrow \nu, \xi - \eta\nabla L(\nu, \xi)$.
25:        Compute intrinsic reward $r_{intr}$ with equation 9.
26:        Compute exploration value loss function $L(\sigma)$ with equation 3.
27:        Update $\sigma \leftarrow \sigma - \eta\nabla L(\sigma)$.
28:        Compute skill value loss function $L(\theta)$ with equation 4.
29:        Compute regularization loss $L_{\text{reg}}(\theta) = \frac{1}{b^2} \sum_{i,j \in I^2} B_\omega(s_i, a_i)$.
30:        Update $\theta \leftarrow \theta - \alpha\nabla L(\theta)$.
31:        Compute actor loss function $L(\zeta)$ with equation 5.
32:        Update $\zeta \leftarrow \zeta - \eta\nabla L(\zeta)$.
33:     **end for**
34:     **/* Update target network parameters */**
35:     $\theta^- \leftarrow \tau\theta + (1 - \tau)\theta^-$
36:     $\omega^- \leftarrow \tau\omega + (1 - \tau)\omega^-$
37:     $\kappa^- \leftarrow \tau\kappa + (1 - \tau)\kappa^-$
38: **end for**=0

---

A.2 FINE-TUNE PSEUDO-CODE

---

**Algorithm 2** DVFB Algorithm: Downstream Fine-tuning Phase

---

1: **Inputs:** replay buffer $\mathcal{D}$, initialized networks $F_\theta$, $B_\omega$, actor network $\pi_\zeta$, downstream task value function $Q_{F,\mu}$, learning rate $\eta$, mini-batch size $b$, number of episodes $E$, number of gradient updates $N$, number of inference step $M$.
2:  /* Inference skill
3: **for** $t = 1$ to $M$ **do**
4:     Sample $z \sim p(z)$
5:     Select action $a_t = \pi_\zeta(s_t, a_t, z)$
6:     Observe next state $s_{t+1}$
7:     Store transition $(s_t, a_t, s_{t+1}, r)$ in the replay buffer $\mathcal{D}$
8: **end for**
9: Inference skill: $z = B(s_t)r$.
10: /* Fine-tune
11: **for** $m = 1$ **do**
12:     /* Collect $E$ episodes
13:     **for** $e = 1$ to $E$ **do**
14:         Observe an initial state $s_0$
15:         **for** $t = 1$ to $T$ **do**
16:             Select action $a_t = \pi_\zeta(s_t, a_t, z)$
17:             Observe next state $s_{t+1}$
18:             Store transition $(s_t, a_t, s_{t+1})$ in the replay buffer $\mathcal{D}$
19:         **end for**
20:     **end for**
21:     /* Perform $N$ stochastic gradient descent updates
22:     **for** $n = 1$ to $N$ **do**
23:         Sample a mini-batch of transitions $\{(s_i, a_i, s_{i+1}, z)\}_{i \in I} \subset \mathcal{D}$ of size $|I| = b$.
24:         Compute downstream task value loss function $L(\mu)$ with equation 10.
25:         Update $\mu \leftarrow \mu - \eta \nabla L(\mu)$.
26:         Compute actor loss function $L(\zeta)$ with equation 11.
27:         Update $\zeta \leftarrow \zeta - \eta \nabla L(\zeta)$.
28:     **end for**
29: **end for**=0

---

## B EXTENDED BACKGROUND

In the presence of extrinsic rewards, Reinforcement Learning (RL) has demonstrated its efficacy in learning powerful task-specific skills (Li et al., 2019; Wu et al., 2022; Tu et al., 2025; Sun et al., 2025). The success of unsupervised learning in computer vision (CV) (Chen et al., 2020) and natural language processing (NLP) (Devlin et al., 2019) inspires researchers and benefits task-specific RL with complex visual input by enhancing representation learning in a target manner. Nevertheless, the task-specific supervision of extrinsic reward makes it difficult for the agents who have been trained with a significant amount of effort to generalize their knowledge to novel tasks (Stooke et al., 2021). In order to enhance this generalization, unsupervised RL is proposed, in which the task-agnostic reward is specifically designed for unsupervised pre-training. The pre-trained feature encoders (Liu & Abbeel, 2021b; Yarats et al., 2021) and exploration policies (Burda et al., 2019; Yarats et al., 2021) can be subsequently implemented to facilitate efficient RL on a variety of downstream tasks. In addition to the generalization, intelligent agents should also be capable of exploring environments and acquiring a variety of useful behaviors without any extrinsic supervision, similar to human beings. For the reasons above, unsupervised skill discovery (Eysenbach et al., 2019; Laskin et al., 2021; Lee et al., 2019) is proposed and becomes a novel research hotspot. As a branch of unsupervised RL, unsupervised skill discovery approaches also design task-agnostic intrinsic rewards and achieve pre-training with these rewards through RL, which guarantees multi-task downstream generalization (Laskin et al., 2022; Liu & Abbeel, 2021a; Liu et al., 2025; Yang et al., 2023; Bai et al., 2024). The primary distinction is that skill discovery necessitates additional input in the form of skill vectors, which are conditions. Their objective is to identify task-agnostic policies that are distinguishable by skill vectors (Park et al., 2024; Liu et al., 2025; Lee et al., 2019). These methodologies demonstrate promising outcomes in a variety of disciplines, including manipulation, video games (Schrittwieser et al., 2020; Ye et al., 2021), robot locomotion (Laskin et al., 2021; Tassa et al., 2018), and so on. However, we observe that existed unsupervised skill discovery methods always lack the ability of zero-shot policy learning, i.e., their learned skills can't directly achieve multi-task adaptation without further task-specific adjustment.

Zero-shot RL methods are typically based on successor representations (Dayan, 1993), universal value function approximators (Schaul et al., 2015), successor features (Barreto et al., 2017), and successor measures. The state-of-the-art methods achieve these ideas either by using universal successor features (USFs) (Schaul et al., 2015) or forward-backward representations (FB) (Touati & Ollivier, 2021; Touati et al., 2023). Methods based on USFs require learning a representation for successor features. Previous approaches have utilized techniques such as Laplacian Eigenfunctions (Wu et al., 2018), Low-Rank Approximation of Transition Probabilities (Ren et al., 2023), Contrastive Learning (Balestriero & LeCun, 2022), and Low-Rank Approximation of Successor Representations (Touati et al., 2023) to learn representations. FB methods directly learn successor measures using forward and backward representations, avoiding the need to learn representations. However, these successor feature-based methods rely on diverse offline datasets, and in online URL, the lack of exploration capabilities results in poor data diversity, thereby affecting the learning of successor measures and zero-shot generalization. This paper proposes a dual-value forward-backward representation framework (DVFB) that enables zero-shot generalization in online URL by encouraging exploration while ensuring the learning of successor features.

## C EXPERIMENTAL SETUP

### C.1 ENVIRONMENTS

Following recent advancements (Liu et al., 2025; Bai et al., 2024), we evaluate task generalization performance on 12 downstream tasks across three domains: *Walker*, *Quadruped*, and *Jaco*, using the URLB (Laskin et al., 2021) and DMC (Tassa et al., 2018) benchmarks. Here are the detailed domains and corresponding tasks:

- The *Walker* domain includes walker control tasks with a state space $S \in \mathbb{R}^{24}$ and action space $A \in \mathbb{R}^6$. This domain includes four tasks: *Walker stand*, *Walker walk*, *Walker flip*, and *Walker run*.

- The *Quadruped* domain involves quadruped control tasks with a higher-dimensional state space $S \in \mathbb{R}^{78}$ and action space $A \in \mathbb{R}^{16}$. It includes the tasks of *Quadruped stand*, *Quadruped walk*, *Quadruped jump*, and *Quadruped run*.

- The *Hopper* domain consists of one-legged hopper control tasks, which present a more challenging exploration of rewards, with a state space $S \in \mathbb{R}^{14}$ and action space $A \in \mathbb{R}^4$. It includes the tasks of *Hopper hop*, *Hopper flip*, *Hopper hop backward*, and *Hopper flip backward*.

In the experiments, each method is pre-trained in the aforementioned environments without rewards for $2e^6$ steps and adapts to downstream tasks through zero-shot for generalization and fine-tuning.

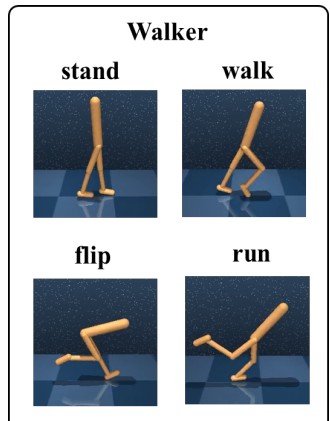 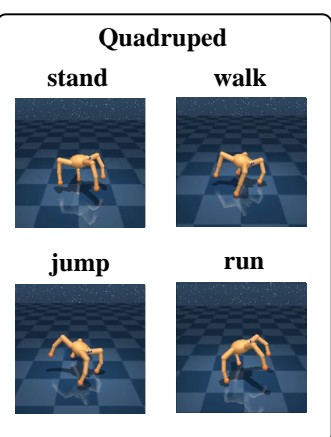 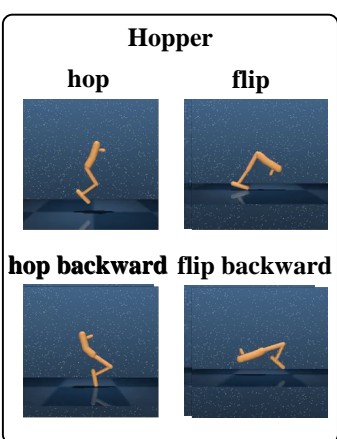

Figure 8: Illustration of domains and downstream tasks in DMC, Each domain has four downstream tasks.

### C.2 DETAILED BASELINES

In zero-shot generalization, we compared DVFB with (i) successor feature methods that perform well in offline settings and (ii) skill discovery methods, including CIC(Laskin et al., 2022), BeCL(Yang et al., 2023), ComSD(Liu et al., 2025), and CeSD(Bai et al., 2024). In successor feature methods, we compared approaches based on contrastive learning (CL)(Balestriero & LeCun, 2022), Low-Rank Approximation of successor representations (LRA-SR)(Touati et al., 2023), Low-Rank Approximation of state transition probabilities (LRA-P)(Ren et al., 2023), and Laplacian Eigenfunctions (Lap)(Wu et al., 2018), along with the FB method(Touati et al., 2023). The successor representation methods learn representations in various ways to compute successor features for generalization, while the FB method achieves generalization by learning successor metrics. The detailed baseline methods with successor feature methods are presented below:

- Successor Feature with Laplacian Eigenfunctions (Lap): Lap focuses on learning the eigenfunctions of the symmetrized MDP graph Laplacian, which is defined through an exploratory policy. It seeks to minimize the difference between feature representations of consecutive states while ensuring that the features constitute an orthonormal basis. Lap

encourages feature clustering among states and their neighbors while effectively separating distant states.

- Low-Rank Approximation of Transition Probabilities (LRA-P): LRA-P approximates the transition probability densities using low-rank models. It learns state-action features by minimizing a contrastive loss that compares samples of consecutive states with independent samples drawn from a stationary distribution. The method produces optimal policies if the model perfectly matches the transition probabilities.

- Contrastive Learning (CL): CL uses a SimCLR-like objective to learn representations by contrasting positive pairs with negative pairs. CL necessitates full trajectory data and is closely related to the spectral decomposition of the successor measure associated with the behavior policy.

- Low-Rank Approximation of Successor Representations (LRA-SR): LRA-SR builds on CL but reduces variance by learning successor measures via temporal difference learning, rather than through Monte Carlo sampling. It normalizes feature representations to form an identity matrix, ensuring low-rank approximation of the successor measure.

- Forward-backward Representations (FB): FB learns successor measure with forward representation and backward representation. It avoids reliance on features, demonstrating outstanding performance in successor feature methods.

In the fine-tuning task adaptation, we compared DVFB with (i) pure exploration methods, including RND(Burda et al., 2019) and Disagreement(Pathak et al., 2019), and (ii) skill discovery methods, including SMM(Lee et al., 2019), DIAYN(Eysenbach et al., 2019), APS(Liu & Abbeel, 2021a), CIC(Laskin et al., 2022), BeCL(Yang et al., 2023), ComSD(Liu et al., 2025), and CeSD(Bai et al., 2024). The detailed baseline methods in task adaptation are as follows:

- Disagreement: Disagreement enhances exploration by training an ensemble of dynamics models, encouraging the agent to maximize the disagreement between them.

- RND: RND is a pure exploration method that encourages the agent to explore unseen states by using a randomly generated network.

- DIAYN: DIAYN maximizes the mutual information (MI) between skills and states by employing a discrete uniform prior to enhance skill entropy. A trainable discriminator is used to estimate state-conditioned entropy, allowing for the computation of intrinsic rewards.

- SMM: SMM aligns the state marginal distribution with a target distribution by formulating a zero-sum game between a state density model and a parametric policy, explicitly maximizing state entropy through intrinsic rewards.

- APS: APS employs another MI decomposition for improved mutual information estimation and utilizes particle-based entropy estimation to enhance exploration.

- CIC: CIC is the first to incorporate contrastive learning into skill discovery, maximizing implicit skill-conditioned entropy by comparing state transitions with skill vectors.

- BeCL: It introduces a novel MI objective to mitigate exploitation issues in CIC, providing theoretical upper bounds.

- ComSD: ComSD employs a contrastive learning-based diversity reward to help agents identify existing skills, combined with a particle-based exploration reward to facilitate the discovery of new behaviors.

- CeSD: CeSD introduces a URL framework that uses an ensemble of skills for partitioned exploration based on state prototypes, enabling local exploration within clusters while maximizing overall state coverage.

While URL methods enable rapid adaptation to downstream tasks through fine-tuning, achieving zero-shot generalization remains difficult due to the absence of a direct link between the learned skills or policies and the downstream tasks. We further demonstrate the properties of baseline method in Table 3.

Table 3: Properties of Baseline Methods

| Method | Zero-shot | Online | Offline | Exploration | Skill Discovery |
|---|---|---|---|---|---|
| Successor Representation Methods | | | | | |
| CL | ✓ | × | ✓ | × | ✓ |
| Lap | ✓ | × | ✓ | × | ✓ |
| LRA-P | ✓ | × | ✓ | × | ✓ |
| LRA-SR | ✓ | × | ✓ | × | ✓ |
| FB | ✓ | × | ✓ | × | ✓ |
| Pure Exploration Methods | | | | | |
| Disagreement | × | ✓ | × | ✓ | × |
| RND | × | ✓ | × | ✓ | × |
| Unsupervised Skill Learning Methods | | | | | |
| DIAYN | × | ✓ | × | ✓ | ✓ |
| SMM | × | ✓ | × | ✓ | ✓ |
| APS | × | ✓ | × | ✓ | ✓ |
| CIC | × | ✓ | × | ✓ | ✓ |
| BeCL | × | ✓ | × | ✓ | ✓ |
| ComSD | × | ✓ | × | ✓ | ✓ |
| CeSD | × | ✓ | × | ✓ | ✓ |
| DVFB(ours) | ✓ | ✓ | × | ✓ | ✓ |

## C.3 DETAILED ZERO-SHOT GENRALIZATION SETUP

We evaluated the zero-shot performance of DVFB across 12 continuous control tasks in the Deep-Mind Control Suite. During the pre-training phase, each method trained for 2 million steps using self-generated data without rewards. Subsequently, each agent interacted for 10,000 steps in the downstream tasks to collect demonstration data, allowing them to acquire the skills most suitable for those tasks. In SR methods, skills $z_r$ are inferred using the formula $r(s) = \phi(s)\top z_r$ , while the FB method infers skills through $z_r = B(s')r$ . The USD methods acquire demonstration data by switching skills in each episode and selects the skill with the highest episode reward as the downstream task skill. During testing, the chosen skills are fixed to evaluate the zero-shot performance of different methods in downstream tasks. Each method was run with 6 random seeds to assess its performance.

During the adaptation to downstream tasks, DVFB and USD methods identify the skills that best match these tasks through skill inference and the selection of the highest-performing skills, respectively. During the fine-tuning phase, USD methods fix the skill vector and trains a new value function using the rewards from downstream tasks, subsequently fine-tuning the skill policy with this updated value function.

## D    HYPER-PARAMETER SETTINGS

In this section, we provide the detailed hyper-parameter settings of our proposed method, as shown in Table 4.

Table 4: Hyper-parameter settings.

| Hyper-parameter | Setting |
|---|---|
| Pre-training frames | $2e^6$ |
| Finetuning frames | $1e5$ |
| Zero-shot selection frames | $1e^4$ |
| RL replay buffer size | $1e6$ |
| Frame stack | 1 |
| Action repeat | 1 |
| Seed frames | 4000 |
| $z$ vector dimensions | 50 |
| $z$ vector space | continuous |
| $z$ update frequency | 300 |
| RL backbone algorithm | DDPG |
| Return discount | 0.99 |
| Discounted steps for return | 3 |
| Batch size | 1024 |
| Optimizer | Adam |
| Learning rate | $1e - 4$ |
| Actor network (MLP) | $\dim(s) + 64 \rightarrow 1024$ $\rightarrow 1024 \rightarrow \dim(a)$ |
| Actor activation | $\text{layernorm(Tanh)} \rightarrow \text{ReLU}$ $\rightarrow \text{Tanh}$ |
| Critic network (MLP) | $\dim(s) + 64 + \dim(a)$ $\rightarrow 1024 \rightarrow 1024 \rightarrow 1$ |
| Actor activation | $\text{layernorm(Tanh)} \rightarrow \text{ReLU}$ |
| Agent update frequency | 2 |
| Target critic network EMA | 0.01 |
| Exploration stddev clip | 0.3 |
| Exploration stddev value | 0.2 |
| coefficient $\alpha$ | 5 |
| coefficient $\beta$ | 0.5 |
| coefficient $\eta$ | 0.5 |

# E    FULL EXPERIMENTS RESULT

As shown in Figure 9, we present the fine-tuning curves of our method compared to eight baselines across all twelve tasks in three domains. Our method demonstrates strong initial performance and achieves further improvements through fine-tuning with greater efficiency.

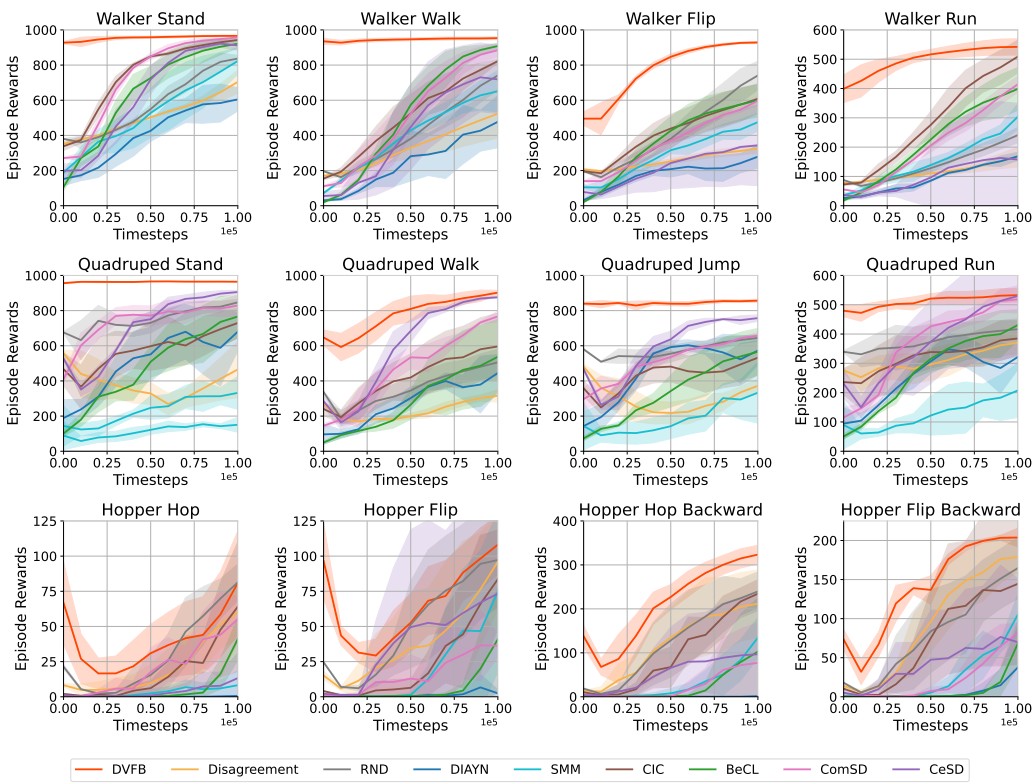

Figure 9: **Finetuning Curve on the *Walker* and *Quadruped* Domains.** DVFB begins with outstanding zero-shot performance and further enhances performance efficiently through fine-tuning.

# F    FULL ABLATION STUDY

Figure 10 presents the ablation study results of DVFB's fine-tuning technique across 12 tasks in three domains. The results demonstrate that the proposed dual-value fine-tuning technique ensures efficient and stable fine-tuning.

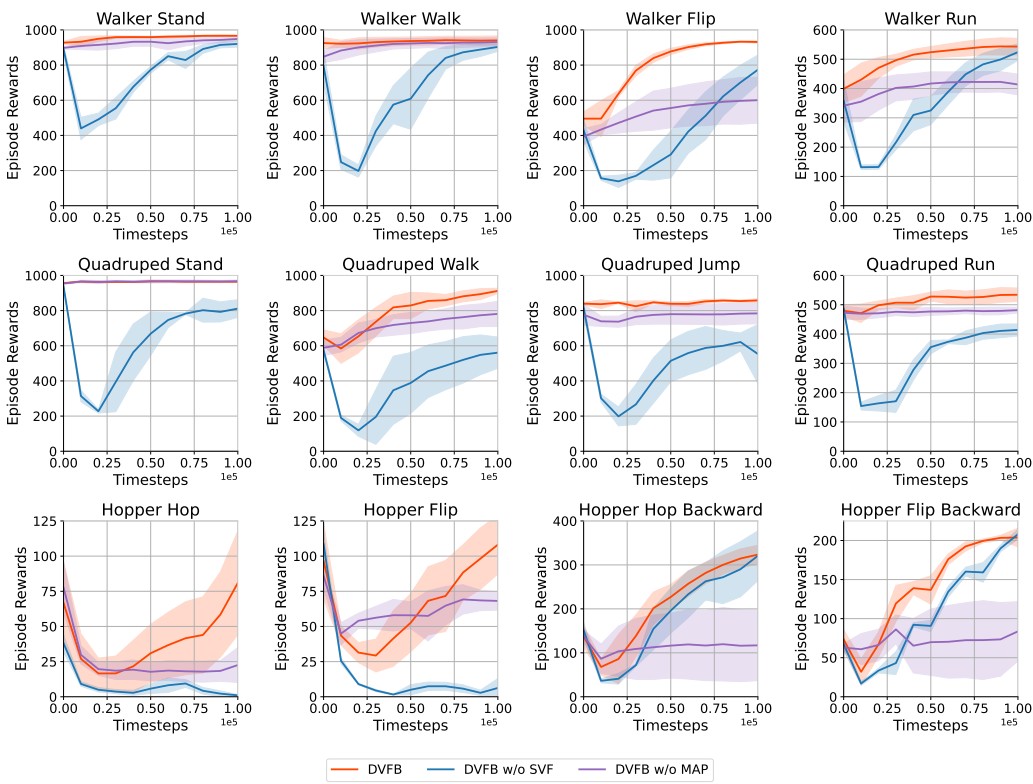

Figure 10: **Ablation study results for the dual-value fine-tuning scheme.** We present the fine-tuning curves for DVFB, DVFB w/o SVF, and DVFB w/o MAP across all 12 downstream tasks. DVFB w/o SVF denotes DVFB fine-tuning with a downstream task value function trained from scratch, while DVFB w/o MAP refers to DVFB fine-tuning without the reward mapping technique in the dual-value scheme.

## G    COMPARISON WITH OFFLINE ZERO-SHOT METHODS

In this section, we compare DVFB against the offline zero-shot methods LAP, LRA-SR, LRA-P and FB in *Walker*, *Quadruped* and *Cheetah* domains. The performances of offline methods are from (Touati et al., 2023), where offline datasets are collected by APS, Proto, and RND exploration methods. The results, summarized in Table 2, highlight the following key differences between offline and online methods:

**High Data Sensitivity for Offline Methods:** Offline methods exhibit significant performance variation depending on the quality of exploration data. On the one hand, the same algorithm requires different exploration datasets in different domains. For example, FB trained with Proto data in the Walker domain achieves best performance (666), while in the Quadruped domain using Proto data yields a huge performance drop (222). On the other hand, different algorithms require different exploration datasets. For example, LRA-P performs best with RND data, while FB performs best with APS data. ***When designing a novel algorithm, how to make sure what kind of exploration dataset is most suitable?*** The intuitive idea is to train the models on different exploration datasets, and compare to find the best performance models. Obviously, it will lead to high computational costs. In contrast, DVFB does not depend on offline datasets, and requires only a single agent pre-training phase to achieve strong zero-shot capability, significantly reducing time and computational overhead. It is simpler and easier to deploy than offline zero-shot methods.

**Performance Limitation on Pre-collected Fixed Dataset:** Offline methods rely on the diversity and quality of fixed pre-collected datasets, which limits their generalization performance. In contrast, DVFB balances the exploration and exploitation online with an intrinsic reward based on contrastive learning, leading to enhanced skill learning and better zero-shot performance. Experimental results across **twelve tasks in Mujoco domains** demonstrate that DVFB consistently outperforms zero-shot offline methods.

In summary, the results demonstrate that DVFB offers superior performance and efficiency compared to both offline and online methods, establishing its significance in zero-shot online URL.

Table 5: Zero-shot Generalization Performance Comparison

| Domain | task | LAP* | | | LRA-P* | | | LRA-SR* | | | Offline FB* | | | DVFB |
|---|---|---|---|---|---|---|---|---|---|---|---|---|---|---|
| | | (APS) | (Proto) | (RND) | (APS) | (Proto) | (RND) | (APS) | (Proto) | (RND) | (APS) | (Proto) | (RND) | |
| Walker | Stand | 895 | 937 | 853 | 643 | 687 | 904 | 591 | 874 | 828 | 822 | 902 | 890 | 905 |
| | Walk | 386 | 883 | 607 | 159 | 300 | 818 | 671 | 867 | 853 | 817 | 917 | 760 | 900 |
| | Flip | 454 | 548 | 569 | 340 | 281 | 512 | 186 | 551 | 454 | 413 | 507 | 578 | 515 |
| | Run | 289 | 280 | 299 | 115 | 183 | 325 | 204 | 391 | 350 | 346 | 336 | 388 | 423 |
| | Average | 506 | **662** | 582 | 314 | 363 | **640** | 413 | **671** | 621 | 600 | **666** | 654 | **686** |
| Quadruped | Stand | 963 | 231 | 720 | 497 | 264 | 552 | 872 | 99 | 944 | 924 | 287 | 815 | 953 |
| | Walk | 524 | 135 | 410 | 228 | 172 | 310 | 463 | 215 | 516 | 712 | 280 | 528 | 624 |
| | Jump | 718 | 177 | 490 | 309 | 184 | 447 | 632 | 134 | 731 | 649 | 183 | 651 | 816 |
| | Run | 491 | 125 | 399 | 238 | 166 | 301 | 448 | 113 | 461 | 476 | 137 | 429 | 467 |
| | Average | **674** | 167 | 505 | 318 | 197 | **403** | 604 | 140 | **663** | 690 | 222 | 606 | **715** |
| Cheetah | run | 198 | 142 | 50 | 8 | 149 | 6 | 247 | 209 | 138 | 276 | 267 | 247 | 271 |
| | run backward | 221 | 146 | 90 | 1 | 133 | 2 | 261 | 230 | 82 | 238 | 238 | 185 | 319 |
| | walk | 900 | 722 | 330 | 75 | 770 | 29 | 918 | 860 | 446 | 844 | 844 | 827 | 906 |
| | walk backward | 937 | 798 | 499 | 8 | 629 | 14 | 983 | 979 | 352 | 981 | 981 | 793 | 978 |
| | Average | **564** | 452 | 242 | 23 | **420** | 13 | **602** | 570 | 255 | **585** | 583 | 513 | **619** |
| All average | | **581** | 427 | 443 | 218 | 326.5 | **352** | **540** | 460 | 513 | **625** | 490 | 591 | **673** |
| Average of best | | | 633 | | | 487 | | | 645 | | | 647 | | **673** |

Results marked with * are sourced from FB(ICLR 23).

# H  THEORETICAL ANALYSIS FOR IMPROVEMENTS OF DVFB

In this section, we offer a theoretical guarantee for zero-shot generalization and a detailed analysis of how the DVFB framework improves FB's zero-shot generalization in online URL.

**Theorem 1.** *(Touati et al., 2023) Let $F : S \times A \times Z \to Z$ and $B : S \times A \to Z$ be successor state approximation functions, and define policy $\pi_z(s) = \arg\max_a F(s, a, z)$ for each $z \in Z$.*

*Given a positive probability distribution $\rho$ on $S \times A$, let $m^\pi$ be the density of successor state measure $M^\pi$ under policy $\pi$ with respect to $\rho$. Define the model estimates:*

$$\hat{m}^\pi(s, a, s', a') := F(s, a, z)^\top B(s', a')$$
$$\hat{M}^\pi(s, a, ds', da') := \hat{m}^\pi(s, a, s', a')\rho(ds', da')$$

*For any bounded reward function $r : S \times A \to \mathbb{R}$, let $V^*$ be the optimal value function, $\bar{V}^{\pi_z}$ be the value function of policy $\pi_z$, and $z_R = \mathbb{E}_{(s,a)\sim\rho}[r(s, a)B(s, a)]$. Then:*

1. *Under bounded density estimation error: If $\mathbb{E}_{(s',a')\sim\rho}|\hat{m}^{z_R}(s, a, s', a') - m^{\pi_z}(s, a, s', a')| \leq \varepsilon$ for all $(s, a) \in S \times A$, then*

$$\|\bar{V}^{\pi_z} - V^*\|_\infty \leq \frac{3\varepsilon\|r\|_\infty}{1 - \gamma}$$

2. *For Lipschitz continuous rewards: If $\|\hat{M}^{z_R}(s, a, \cdot) - M^{\pi_z}(s, a, \cdot)\|_{\mathrm{KR}} \leq \varepsilon$ for all $(s, a) \in S \times A$, then*

$$\|\bar{V}^{\pi_z} - V^*\|_\infty \leq \frac{3\varepsilon \max(\|r\|_\infty, \|r\|_{\mathrm{Lip}})}{1 - \gamma}$$

3. *For general norm pairs: Given norms $\|\cdot\|_A$ on functions and $\|\cdot\|_B$ on measures satisfying $\int f d\mu \leq \|f\|_A\|\mu\|_B$, for any reward function with $\|r\|_A < \infty$,*

$$\|\bar{V}^{\pi_z} - V^*\|_\infty \leq \frac{3\|r\|_A}{1 - \gamma}\sup_{s,a}\|\hat{M}^{z_R}(s, a, \cdot) - M^{\pi_z}(s, a, \cdot)\|_B$$

*Furthermore, the Q-function approximation satisfies:*

$$\sup_{s,a}|F(s, a, z_R)^\top z_R - Q^*(s, a)| \leq \frac{2\|r\|_A}{1 - \gamma}\sup_{s,a}\|\hat{M}^{z_R}(s, a, \cdot) - M^{\pi_z}(s, a, \cdot)\|_B$$

The Theorem 1 establishes that for any reward function $r$, the error in value estimation is bounded as follows:

$$\sup_{S\times A}|Q^{\pi_{z_R}} - Q^*| \leq \frac{3|r|A}{1 - \gamma}\sup_{S\times A}|\varepsilon_{z_R}(s, a, \cdot)|B, \tag{13}$$

where $\varepsilon_{z_R}(s, a, \cdot) = M^{\pi_z}(s, a, ds', da') - F(s, a, z)^\top B(s', a')\rho(ds', da')$ represents the difference between the true successor measure and the estimated successor measure. This bound ensures that the zero-shot generalization error depends on the accuracy of the successor measure estimation.

**Theoretical Analysis for Improvements of DVFB.** In online URL, the performance of the Forward-Backward (FB) method is hindered by insufficient exploration, leading to inaccurate estimation of successor measures. Our empirical analysis in Section 4 highlights this limitation by showing the poor correlation between the value function and returns on downstream tasks under FB. To address this, we propose the Dual-Value Forward-Backward (DVFB) framework, which introduces a dual-value structure to enhance exploration and improve successor measure estimation. Specifically, let $\varepsilon_{z_R}^{DVFB}$ and $\varepsilon_{z_R}^{FB}$ represent the successor measure errors under DVFB and FB, respectively. DVFB reduces these errors, such that:

$$|\varepsilon_{z_R}^{DVFB}|B \leq |\varepsilon_{z_R}^{FB}|B.$$

As a result, for any unseen reward function $r_{\text{new}}$, the zero-shot generalization error of DVFB is bounded by:

$$|Q^{\pi_{\text{DVFB}}}_{r_{\text{new}}} - Q^*_{r_{\text{new}}}| \leq \frac{3|r_{\text{new}}|A}{1-\gamma}|\varepsilon^{DVFB}_{z_R}|B \leq \frac{3|r_{\text{new}}|A}{1-\gamma}|\varepsilon^{FB}_{z_R}|B.$$

This tighter bound demonstrates that DVFB achieves superior zero-shot generalization by reducing successor measure estimation errors via enhanced exploration.

Our experimental results validate this theoretical analysis, with the DVFB framework consistently outperforming FB across a range of unseen tasks. These findings confirm that DVFB offers improved zero-shot generalization in online URL.

# I FURTHER EXPERIMENTAL RESULTS ON NAVIGATION AND ROBOTIC MANIPULATION DOMAIN

we have conducted additional experiments on two distinct domains: a Point-Mass Maze navigation environment and a Meta-World robotic manipulation environment. Although the offline methods FB-offline(Touati et al., 2023) and MCFB-offline [2] rely on offline data and offline settings (in contrast to online URL, which requires multi-stage training and more expensive computation), we provide a comparison with these offline reinforcement learning methods for a more comprehensive evaluation of our approach. As they didn't do the Meta-World experiment, we summarize the results of FB-offline and MCFB-offline with RND offline data in Point-Mass Maze domain. As shown in Table 6, DVFB demonstrates better performance across both domains. The additional results demonstrate that DVFB is not only effective in robotic control tasks but also generalizes well to other domains, such as navigation and robotic manipulation.

Table 6: Performance comparison across different domains. For Point-Mass Maze, results show mean ± standard deviation across three seeds. For Meta-World, results show success rates.

| Domain | Task | FB | CIC | CeSD | FB-offline* | MCFB-offline* | DVFB |
|---|---|---|---|---|---|---|---|
| Point-Mass | Reach Top-left | 69 ± 6 | 18 ± 6 | 12 ± 8 | 612 | 773 | **932 ± 10** |
| | Reach Top-right | 77 ± 95 | 5 ± 2 | 5 ± 4 | 0 | **270** | 203 ± 81 |
| | Reach Bottom-left | 3 ± 3 | 7 ± 4 | 18 ± 21 | **268** | 1 | 94 ± 45 |
| | Reach Bottom-right | 0 ± 0 | 2 ± 2 | 2 ± 2 | 0 | 0 | **4 ± 3** |
| | Average | 37.3 | 8.0 | 9.3 | 219 | 261 | **308.3** |
| Meta-World | Faucet Open | 0.18 | 0.04 | 0.00 | — | — | **0.60** |
| | Faucet Close | 0.10 | 0.18 | 0.00 | — | — | **0.52** |

Results marked with * are sourced from MCFB(Jeen et al., 2024).

## J    IMPORTANCE OF CONTRASTIVE ENTROPY REWARD

In this section, we discuss the importance of the contrastive entropy reward. The contrastive entropy reward is designed to promote skill discrimination during exploration, thereby preserving the skills learned through the FB mechanism. To assess the role of the contrastive entropy reward, we conducted two sets of comprehensive experiments.

**Ablation Study.** We perform an ablation study on the coefficient $\beta$ of the contrastive entropy reward, as shown in Table 7. The results demonstrate that increasing $\beta$ from 0.1 to 0.7 consistently improves performance, validating that the contrastive entropy reward enhances generalization by promoting skill separability.

Table 7: Ablation study on contrastive entropy coefficient $\beta$ on Walker tasks. Results show mean ± standard deviation across three seeds.

| Task | $\beta$=0.1 | $\beta$=0.3 | $\beta$=0.5 | $\beta$=0.7 | $\beta$=0.9 |
|---|---|---|---|---|---|
| Stand | 819±32 | 862±9 | 905 | 898±62 | 919±9 |
| Walk | 819±38 | 861±18 | 900 | 926±17 | 873±32 |
| Flip | 428±10 | 501±18 | 515 | 616±129 | 453±30 |
| Run | 344±28 | 397±40 | 423 | 434±54 | 342±35 |
| Average | 603 | 655 | 686 | **719** | 647 |

**Comparison Experiment.** We compare DVFB with variants that use alternative intrinsic rewards (ICM-APT, Proto, and CIC), as shown in Table 8.

Table 8: Comparison with alternative intrinsic rewards on Walker tasks. Results show mean ± standard deviation across three seeds.

| Task | DVFB(ICM-APT) | DVFB(Proto) | DVFB(CIC) | DVFB |
|---|---|---|---|---|
| Stand | 883±106 | 844±101 | 846±74 | **905±27** |
| Walk | 840±85 | 821±27 | 825±24 | **900±53** |
| Flip | 436±68 | 454±51 | 436±140 | **515±67** |
| Run | 354±15 | 358±17 | 342±14 | **423±53** |
| Average | 628 | 619 | 612 | **686** |

The results reveal several key insights: (1) **The scalability of DVFB.** All variants achieve reasonable zero-shot generalization performance, demonstrating DVFB's compatibility with different intrinsic rewards. (2) **The advantage of the CE reward.** DVFB with contrastive entropy consistently outperforms the other variants, achieving the highest average performance.

These experiments provide strong evidence for the effectiveness of our contrastive entropy design and demonstrate DVFB's flexibility in incorporating different intrinsic rewards.

## K  IMPORTANCE OF REWARD MAPPING TECHNIQUE

In this section, we provide a detailed justification and explore potential alternatives for reward mapping technique during our experiments.

**Reward Mapping Implementation.** The reward mapping process involves three key steps: (1) Compute the implicit reward using the backward network (backward_net) and a latent variable (z). (2) Normalize both implicit and extrinsic rewards using running mean and standard deviation trackers. (3) Rescale the extrinsic reward to align with the scale of the implicit reward.

The detailed implementation is shown in the following pseudocode:

---
**Algorithm 3** Reward Mapping Mechanism

---
1: **Input:** downstream reward $r_t$, next state $s_{\text{next}}$, backward representation network $B_{\text{net}}$, skill $z$, implicit reward normalization function $r_{fb}\_rms$, downstream reward normalization function $r_t\_rms$.
1: **procedure** REWARD MAPPING
2: /* Compute features from backward network
2:     $B \leftarrow B_{\text{net}}(s_{\text{next}})$
3: /* Compute Covariance matrix
3:     $\Sigma \leftarrow \frac{1}{m} B^{\top} B$
3:     $\Sigma^{-1} \leftarrow \text{inverse}(\Sigma)$
4: /* Compute Implicit reward
4:     $r_{\text{implicit}} \leftarrow \sum_{i=1}^{m} \left( B_i \cdot \Sigma^{-1} \cdot z \right)$
5: /* Update normalization functions
5:     $(\mu_{\text{fb}}, \sigma_{\text{fb}}^2) \leftarrow \text{fb\_rms.update}(r_{\text{implicit}})$
5:     $(\mu_{\text{t}}, \sigma_{\text{t}}^2) \leftarrow \text{t\_rms.update}(r_{\text{t}})$
6: /* Rescale reward
6:     $r_{\text{rescaled}} \leftarrow r_{\text{t}} \cdot \frac{\mu_{\text{fb}}}{\mu_{\text{t}}+\epsilon}$
6:     **return** $r_{\text{rescaled}}$
6: **end procedure**=0

---

**Potential Fine-tuning Techniques.** The goal of fine-tuning is to ensure stable policy improvements, guided by both the prior skill value $Q_M$ and the downstream task value $Q_F$. The reward mapping technique is designed to balance the influence of these two values. A straightforward alternative is to directly use the downstream task rewards $r_t$ for the task value, adjusting the coefficient $\eta$ in Eq. 11 to balance the importance of $Q_M$ and $Q_F$. We refer to this as DVFB w/o MAP, which employs a dual-value fine-tuning scheme based on downstream task rewards $r_t$ rather than $r_f$. Additionally, we explore an adaptive approach where the $\eta$ parameter is dynamically adjusted as $\eta = \frac{Q}{MQ}$, which we call DVFB w/o MAP adaptive.

Table 9: Performance comparison of different fine-tuning approaches on the quadruped domain. Results show mean±std over three runs.

| Task | DVFB w/o MAP | | | | DVFB |
|---|---|---|---|---|---|
| | $\eta = 0.02$ | $\eta = 0.1$ | $\eta = 0.5$ | adaptive $\eta$ | |
| Stand | $954 \pm 5$ | $951 \pm 10$ | $961 \pm 8$ | $960 \pm 5$ | $965 \pm 7$ |
| Walk | $753 \pm 32$ | $765 \pm 18$ | $752 \pm 31$ | $820 \pm 77$ | $908 \pm 21$ |
| Jump | $819 \pm 28$ | $811 \pm 16$ | $784 \pm 59$ | $830 \pm 12$ | $831 \pm 20$ |
| Run | $496 \pm 23$ | $491 \pm 4$ | $490 \pm 14$ | $496 \pm 48$ | $536 \pm 27$ |
| Average | $756$ | $755$ | $747$ | $777$ | $804$ |

**Experimental Results.** We perform comparative experiments in the quadruped domain, with results presented in Figure 11 and Table 9. Our findings show that DVFB w/o MAP provides stable but limited improvements with higher coefficients, while performance becomes unstable with lower coefficients, ultimately limiting overall performance. Moreover, DVFB w/o MAP adaptive struggles to achieve superior improvements due to the nonlinear relationship between skill value and

downstream value. In contrast, DVFB with the reward mapping scheme, which uses all coefficients, consistently delivers stable and superior improvements across all tasks. These results strongly validate the effectiveness of our chosen approach.

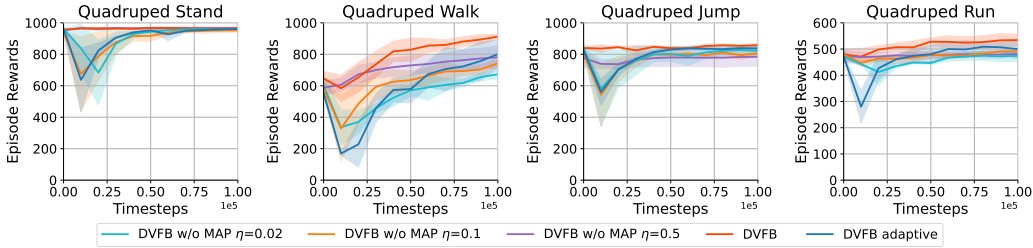

Figure 11: **Finetuning Curve on the *Quadruped* Domains for DVFB with different fine-tuning approaches.**

## L    THE SENSITIVITY OF DVFB TO HYPERPARAMETERS

In this section, we conduct a series of ablation studies to evaluate the impact of the key hyperparameters $\alpha$, $\beta$ and $\eta$ on DVFB's performance. Below, we present the experimental results.

**Ablation Study on $\alpha$.** Table 10 shows the results for varying $\alpha$ in Walker domain. The experiments indicate that while changes in $\alpha$ affect performance slightly, the overall generalization performance of DVFB remains stable, with the best results observed at $\alpha = 5$.

Table 10: Performance of DVFB with different $\alpha$ values on the Walker domain. Results show mean ± standard deviation across three seeds.

| **Task** | $\alpha = 1$ | $\alpha = 3$ | $\alpha = 5$ | $\alpha = 7$ | $\alpha = 9$ |
|---|---|---|---|---|---|
| Stand | $911 \pm 5$ | $912 \pm 3$ | $905 \pm 27$ | $888 \pm 5$ | $807 \pm 33$ |
| Walk | $835 \pm 71$ | $895 \pm 41$ | $900 \pm 53$ | $862 \pm 7$ | $707 \pm 49$ |
| Flip | $464 \pm 76$ | $522 \pm 92$ | $515 \pm 67$ | $489 \pm 18$ | $423 \pm 19$ |
| Run | $350 \pm 69$ | $444 \pm 13$ | $423 \pm 53$ | $345 \pm 5$ | $266 \pm 45$ |
| **Average** | 640 | **693** | 686 | 646 | 551 |

**Ablation Study on $\beta$.** Similarly, Table 11 reports the performance for different $\beta$ values in Walker domain. The results reveal that DVFB achieves optimal performance at $\beta = 0.7$, with only minor deviations observed across other values. This demonstrates the robustness of our framework to changes in $\beta$.

Table 11: Performance of DVFB with different $\beta$ values on the Walker domain. Results show mean ± standard deviation across three seeds.

| **Task** | $\beta = 0.1$ | $\beta = 0.3$ | $\beta = 0.5$ | $\beta = 0.7$ | $\beta = 0.9$ |
|---|---|---|---|---|---|
| Stand | $819 \pm 32$ | $862 \pm 9$ | $905 \pm 27$ | $898 \pm 62$ | $919 \pm 9$ |
| Walk | $819 \pm 38$ | $861 \pm 18$ | $900 \pm 53$ | $926 \pm 17$ | $873 \pm 32$ |
| Flip | $428 \pm 10$ | $501 \pm 18$ | $515 \pm 67$ | $616 \pm 129$ | $453 \pm 30$ |
| Run | $344 \pm 28$ | $397 \pm 40$ | $423 \pm 53$ | $434 \pm 54$ | $342 \pm 35$ |
| **Average** | 603 | 655 | 686 | **719** | 647 |

**Ablation Study on $\eta$.** We conduct an ablation study in Quadruped domain to evaluate the sensitivity of the DVFB framework to the hyperparameter $\eta$ during the fine-tuning phase. As shown in Table 12, the overall average performance across all tasks remains consistent, indicating that DVFB is relatively resilient to changes in $\eta$.

Table 12: Performance of DVFB with different $\eta$ values on the Quadruped domain. Results show mean ± standard deviation across three seeds.

| **Task** | $\eta = 0.02$ | $\eta = 0.1$ | $\eta = 0.5$ | $\eta = 1.0$ |
|---|---|---|---|---|
| Stand | $957\pm4$ | $964\pm6$ | $965\pm7$ | $954\pm10$ |
| Walk | $891\pm32$ | $908\pm30$ | $908\pm21$ | $886\pm8$ |
| Jump | $830\pm18$ | $838\pm11$ | $831\pm20$ | $835\pm8$ |
| Run | $557\pm15$ | $530\pm15$ | $536\pm27$ | $543\pm26$ |
| **Average** | 809 | 810 | 810 | 804 |

The results of our sensitivity analysis demonstrate that DVFB is resilient to variations in key hyperparameters $\alpha$, $\beta$ and $\eta$. While some fluctuations in performance occur, the overall generalization ability remains largely unaffected, indicating the robustness of our framework across different parameter settings. Furthermore, for all other neural network hyperparameters (e.g., learning rate), we adopt the default settings of URL with DDPG, ensuring consistency with prior work.

