# OpenReview forum: "Unsupervised Zero-Shot Reinforcement Learning via Dual-Value Forward-Backward Representation"
_ICLR.cc/2025/Conference — ICLR 2025 Poster_

### Official Review · Reviewer_pWXy · 2024-11-02

**Soundness:** 3
**Presentation:** 3
**Contribution:** 3
**Rating:** 6
**Confidence:** 5

**Summary:**

The study presents the Dual-Value Forward-Backward (DVFB) framework for zero-shot generalization in online unsupervised reinforcement learning (URL). DVFB integrates a skill value function with an exploration value function to enhance data diversity and generalization in the absence of task-specific rewards. It utilizes a contrastive entropy intrinsic reward to improve exploration and a dual-value fine-tuning method to optimize downstream task performance, claiming good results in continuous control tasks.

**Strengths:**

1. The research presents a novel Dual-Value Forward-Backward (DVFB) paradigm that integrates skill and exploratory value functions to improve data variety and zero-shot generalization in online URL, providing an innovative method for reward-free learning.

2. Should the suggested DVFB approach demonstrate efficacy, it may rectify a basic constraint in reinforcement learning by facilitating zero-shot generalization absent task-specific incentives, hence potentially enabling RL agents to adapt more readily to varied real-world contexts.

**Weaknesses:**

1. The experimental configuration and illustrations are challenging to interpret, with scant explanation offered for particular measures and comparisons. Enhanced labeling, elucidation of axes and benchmarks, and uniform layout throughout figures would facilitate comprehension of the data and augment the paper's readability. Figure 6 has mixed x-axis labels, which needs an improvement. Legends can be bigger w/o affecting the size of total figure for example Figure 7.

2. The method depends on several essential network hyperparameters given in Table-3 yet the research fails to analyze the sensitivity of the results to these selections. An investigation of network hyperparameter sensitivity would enhance confidence in the robustness and generalizability of the findings.

3.The implementation and/or utilization of the reward mapping technique for fine-tuning can be clarified. Integrating pseudocode would improve the accessibility and reproducibility of this component.

4. The report omits a discussion of potential limitations, including computing cost, scalability, and difficulty in real-world implementation. Recognizing these factors might yield a more equitable viewpoint and inform subsequent research.

**Questions:**

1. Could you furnish more detailed explanations regarding the metrics employed in the studies, especially for figures where the axes and comparisons lack clarity? Supplementary labeling and contextual information would assist readers in appropriately interpreting your findings.

2. What is the performance of DVFB in relation to other contemporary zero-shot generalization methods, and what are the reasons for the selection or exclusion of specific baselines? Incorporating a broader array of comparisons or elaborating on these selections would bolster the assertion of enhanced performance.

3. Could you provide a detailed explanation of the practical execution of the reward mapping technique, possibly including pseudocode? Additional detail would elucidate this component's impact during fine-tuning.

4. In what manner does the contrastive entropy reward facilitate skill differentiation, and can you present empirical data that substantiates its efficacy? An elucidation or ablation of the role of this reward would improve comprehension.

5. Have you performed any analysis to assess the sensitivity of DVFB to essential hyperparameters? This would be beneficial to evaluate the resilience of the framework across diverse contexts and circumstances.

6. Could you elaborate on the possible limits of DVFB, including computational complexity and scalability in practical applications? Considering these criteria would yield a more equitable perspective on the approach's practical viability.

---

> ### Author Response · Authors · 2024-11-22
> **Author response to review by reviewer pWXy**
>
> Thank you for your thoughtful and detailed feedback, which will greatly help improve our work. Below are our detailed responses.
>
> ### **Q1. Could you furnish more detailed explanations regarding the metrics employed in the studies, especially for figures where the axes and comparisons lack clarity? Supplementary labeling and contextual information would assist readers in appropriately interpreting your findings.**
>
> **A1.** Thank you for your detailed feedback. We agree that these suggestions will enhance the clarity and readability of the paper. In response, we have added more detailed explanations for the figures and metrics used in the studies. Specifically, we have standardized the x-axis labels in Figure 6 and enlarged the legend fonts in Figure 7, while ensuring that the overall figure dimensions remain consistent. These adjustments aim to make the figures and their comparisons easier to interpret for readers.
>
> ---
>
> ### **Q3. Could you provide a detailed explanation of the practical execution of the reward mapping technique, possibly including pseudocode? Additional detail would elucidate this component's impact during fine-tuning.**
>
> **A3.** The reward mapping technique in our framework is designed to balance the influence of the skill value function with the downstream task value function, ensuring stable and effective fine-tuning. Below, we provide a detailed explanation of this technique and demonstrate its impact through additional experiments.
>
> **Reward Mapping Implementation.**
> The reward mapping process involves three key steps:
> 1. Compute the implicit reward using the backward network (backward_net) and a latent variable (z).
> 2. Normalize both implicit and extrinsic rewards using running mean and standard deviation trackers.
> 3. Rescale the extrinsic reward to align with the scale of the implicit reward.
>
> The detailed implementation and pseudocode is shown in Appendix K (also see https://sites.google.com/view/rewardmapping).
>
> **Potential Fine-Tuning Techniques.**
> For fine-tuning, our objective is to ensure stable policy improvements over zero-shot performance. The reward mapping is designed to achieve stable improvement by balancing the importance of prior skill value $Q_M$ and downstream task value $Q_F$. A straightforward alternative is directly using the downstream task rewards $r_t$ for the task value, and choosing a suitable coefficient $\eta$ in Eq.11 to balance the value of $Q_M$ and $Q_F$. We call this approach **DVFB w/o MAP**, using a dual-value fine-tuning scheme with the downstream task value based on downstream task rewards $r_t$ rather than $r_f$. We also explore an adaptive method to dynamically adjust the $\eta$ parameter by setting $\eta = \frac{Q}{MQ}$, which we refer to as **DVFB w/o MAP adaptive**.
>
> **Experimental Results.**
> We conducted comparative experiments in the quadruped domain, with the results presented in Appendix K (also see https://sites.google.com/view/figureeta) and Table 2. The findings show that DVFB w/o MAP achieves stable but limited improvement with higher coefficients, while exhibiting unstable performance with lower coefficients, ultimately constraining overall performance. It is also challenging for DVFB w/o MAP adaptive to achieve superior improvement due to the nonlinear relationship between skill value and downstream value. In contrast, DVFB with the reward mapping technique consistently delivers stable and superior improvements across all tasks. These results validate the effectiveness of our chosen approach.
>
> **Table 2 Performance Comparison of Different Fine-Tuning Approaches**
> | Task  | DVFB w/o MAP ($\eta = 0.02$) | DVFB w/o MAP ($\eta = 0.1$) | DVFB w/o MAP ($\eta = 0.5$) | DVFB w/o MAP (adaptive $\eta$) | DVFB |
> |-------|-----------------------------|-----------------------------|-----------------------------|-----------------------------|-------|
> | Stand | 954 $\pm$ 5                 | 951 $\pm$ 10                | 961 $\pm$ 8                 | 960 $\pm$ 5                 | 965 $\pm$ 7 |
> | Walk  | 753 $\pm$ 32                | 765 $\pm$ 18                | 752 $\pm$ 31                | 820 $\pm$ 77                | 908 $\pm$ 21 |
> | Jump  | 819 $\pm$ 28                | 811 $\pm$ 16                | 784 $\pm$ 59                | 830 $\pm$ 12                | 831 $\pm$ 20 |
> | Run   | 496 $\pm$ 23                | 491 $\pm$ 4                 | 490 $\pm$ 14                | 496 $\pm$ 48                | 536 $\pm$ 27 |
> | **Average** | 756              | 755                     | 747                     | 777                     | **804** |
>
>
> **Justification for the Reward Mapping Technique.**
> The reward mapping mechanism in DVFB helps balance the prior skill value and downstream task value, enabling stable and effective fine-tuning. Our experiments demonstrate that this approach significantly outperforms alternatives.

---

> ### Author Response · Authors · 2024-11-22
> **Continued**
>
> ### **Q2. What is the performance of DVFB in relation to other contemporary zero-shot generalization methods, and what are the reasons for the selection or exclusion of specific baselines? Incorporating a broader array of comparisons or elaborating on these selections would bolster the assertion of enhanced performance.**
>
> **A2.** We appreciate the reviewer's valuable suggestion regarding the comparison with contemporary zero-shot methods. In this paper, we address zero-shot generalization in online URL and aim to develop a method that combines both zero-shot generalization capability and fine-tuning ability. To the best of our knowledge, no existing method achieves zero-shot generalization in online URL. Therefore, we choose the baselines as they represent the closest existing approaches to our problem setting (**zero-shot generalization in online URL**).
>
> **USD methods** (e.g., BeCL, CeSD) are designed for online unsupervised pre-training and aim to learn transferable skills. While zero-shot generalization would be ideal, they typically rely on fine-tuning in downstream tasks due to limitations in mutual information-based skill learning[1]. Our results in Table 1 demonstrate that DVFB achieves zero-shot performance comparable to their fine-tuned performance, and DVFB outperforms them in fine-tuning.
>
> **Table 1 Comparison of Zero-shot and Fine-tune Performance**
> | Domain | Zero shot | BeCL | CeSD | DVFB | Fine-tune | BeCL* | CeSD* | DVFB |
> |--------|-----------|------|------|------|-----------|-------|-------|------|
> | Walker (average) |     | 59 | 113 | **686** |   | 708 | 668 | **852** |
> | Quadruped (average) |    | 103 | 375 | **715** |    | 720 | 787 | **804** |
> | Hopper (average) |     | 1 | 3 | **101** |   | 20 | 62 | **163** |
> *Results marked with * are sourced from BeCL (ICML 23) and CeSD (ICML 24).*
>
> **SR methods** (e.g., LRA-SR,FB), while originally designed for offline settings, represent SOTA in zero-shot generalization. To ensure a fair comparison of zero-shot capabilities, we implemented their online versions following the authors' official code.
>
> Our experimental setup ensures fair comparison by evaluating methods under identical conditions (same interaction and fine-tuning steps), demonstrating DVFB's superior performance in both zero-shot and fine-tuning scenarios.
>
> **Baseline Categories.** To make it easier to understand the key properties of each method and their relationships, we categorize the methods into several groups based on their key properties as shown in Appendix C.2 (also see https://sites.google.com/view/baselinestable).
>
> **Comparison with offline settings.** Original offline SR methods rely on pre-collected datasets, which diverges from our online URL setup. Hence, such methods were excluded from our primary comparisons. However, we conduct further experiments to provide a broader understanding of the relative advantages of DVFB. We compare DVFB against SR methods, including LAP, LRA-SR, LRA-P, and FB, across the three domains under offline settings. The performances of offline methods are from FB [2], where offline datasets are collected by APS, Proto, and RND exploration methods. The results, summarized in Appendix G (also see https://sites.google.com/view/zero-shot-generalization-perfo), highlight the key differences between offline and online methods:
>
> **High data sensitivity for offline methods:** Offline methods exhibit significant performance variation depending on the quality of exploration data. On the one hand, the same algorithm requires different exploration datasets in different domains. For example, FB trained with Proto data in the Walker domain achieves best performance (666), while in the Quadruped domain using Proto data yields a huge performance drop (222). On the other hand, different algorithms require different exploration datasets. For example, LRA-P performs best with RND data, while FB performs best with APS data. **When designing a novel algorithm, how to make sure what kind of exploration dataset is most suitable?** The intuitive idea is to train the models on different exploration datasets, and compare to find the best performance models. Obviously, it will lead to high computational costs. In contrast, DVFB does not depend on offline datasets, and requires only a single agent pre-training phase to achieve zero-shot capability, significantly reducing time and computational overhead.
>
> **Performance limitation on pre-collected fixed datasets:** Offline methods rely on the diversity and quality of fixed pre-collected datasets, which limits their generalization performance. In contrast, DVFB balances the exploration and exploitation online with an intrinsic reward based on contrastive learning, leading to enhanced skill learning and better zero-shot performance.
>
> In summary, the results demonstrate that DVFB offers superior performance and efficiency compared to both offline and online methods, establishing its significance in zero-shot online URL.

---

> ### Author Response · Authors · 2024-11-22
> **Continued**
>
> ### **Q4. In what manner does the contrastive entropy reward facilitate skill differentiation, and can you present empirical data that substantiates its efficacy? An elucidation or ablation of the role of this reward would improve comprehension.**
>
> **A4.** We agree that a deeper analysis of contrastive entropy reward would strengthen our work.
>
> **Contrastive Entropy Reward.**
> Our contrastive entropy reward is specifically designed to encourage exploration while keeping skill separability through two mechanisms:
>
> - **Skill Discriminator.** We employ contrastive learning to train a skill discriminator that learns representation for trajectories and skills. The dot product between trajectory representation and skill representation serves as a similarity measure. The learned discriminator is used to distinguish skill-trajectory pairs.
> - **Contrastive Entropy.** We maximize a particle entropy computed from the dissimilarity between a trajectory and its N most related skills. This encourages the agent to keep skill separability while exploring new states.
>
> **Ablation Study.**
> We perform an ablation study on the coefficient \(\beta\) of contrastive entropy reward \(r_{intr} = r_{rnd} + \beta r_{ce}\), as shown in Table 3. The results demonstrate that increasing \(\beta\) from 0.1 to 0.7 consistently improves performance, validating that contrastive entropy enhances generalization by promoting skill separability. However, further increasing \(\beta\) negatively impacts performance due to an imbalance between skill learning and exploration.
>
> **Table 3 Ablation study on contrastive entropy coefficient $\beta$ on Walker tasks. Results show mean ± standard deviation across three seeds.**
> | Task | \(\beta = 0.1\) | \(\beta = 0.3\) | \(\beta = 0.5\) | \(\beta = 0.7\) | \(\beta = 0.9\) |
> |------|-----------------|-----------------|-----------------|-----------------|-----------------|
> | Stand | 819±32          | 862±9           | 905             | 898±62          | 919±9           |
> | Walk  | 819±38          | 861±18          | 900             | 926±17          | 873±32          |
> | Flip  | 428±10          | 501±18          | 515             | 616±129         | 453±30          |
> | Run   | 344±28          | 397±40          | 423             | 434±54          | 342±35          |
> | **Average** | 603         | 655         | 686         | **719**         | 647         |
>
>  **Comparison Experiment.**
> We further compare DVFB with its variants using alternative intrinsic rewards (ICM-APT, Proto, and CIC), as presented in Table 4.
>
> **Table 4 Comparison with alternative intrinsic rewards on Walker tasks. Results show mean ± standard deviation across three seeds.**
> | Task  | DVFB (ICM-APT)  | DVFB (Proto)    | DVFB (CIC)      | DVFB            |
> |-------|-----------------|-----------------|-----------------|-----------------|
> | Stand | 883±106         | 844±101         | 846±74          | **905±27**      |
> | Walk  | 840±85          | 821±27          | 825±24          | **900±53**      |
> | Flip  | 436±68          | 454±51          | 436±140         | **515±67**      |
> | Run   | 354±15          | 358±17          | 342±14          | **423±53**      |
> | **Average** | 628         | 619         | 612         | **686**         |
>
> The results reveal several key insights:
> 1. **The scalability of DVFB.** All variants achieve reasonable zero-shot generalization performance, demonstrating DVFB's compatibility with different intrinsic rewards.
> 2. **The advantage of CE reward.** DVFB with contrastive entropy consistently outperforms other variants, achieving the highest average performance.
>
> These experiments provide evidence for the effectiveness of our contrastive entropy design, while also demonstrating DVFB's flexibility in incorporating different intrinsic rewards.

---

> ### Author Response · Authors · 2024-11-22
> **Continued**
>
> ### **Q5. Have you performed any analysis to assess the sensitivity of DVFB to essential hyperparameters? This would be beneficial to evaluate the resilience of the framework across diverse contexts and circumstances.**
>
> **A5.** Thank you for highlighting the importance of hyperparameter sensitivity analysis. To address this concern, we conduct a series of ablation studies to evaluate the impact of the key hyperparameters $ \alpha $, $ \beta $, and $ \eta $ on DVFB’s performance.
>
> **Ablation Study on $ \alpha $.**
> Table 5 shows the results for varying $ \alpha $ in the Walker domain. The experiments indicate that while changes in $ \alpha $ affect performance slightly, the overall generalization performance of DVFB remains stable.
>
> **Table 5 Ablation Study on $ \alpha $**
> | **Task** | $ \boldsymbol{\alpha = 1} $ | $ \boldsymbol{\alpha = 3} $ | $ \boldsymbol{\alpha = 5} $ | $ \boldsymbol{\alpha = 7} $ | $ \boldsymbol{\alpha = 9} $ |
> |----------|-----------------------------|-----------------------------|-----------------------------|-----------------------------|-----------------------------|
> | Stand    | 911 $\pm$ 5                 | 912 $\pm$ 3                 | 905 $\pm$ 27                | 888 $\pm$ 5                 | 807 $\pm$ 33                |
> | Walk     | 835 $\pm$ 71                | 895 $\pm$ 41                | 900 $\pm$ 53                | 862 $\pm$ 7                 | 707 $\pm$ 49                |
> | Flip     | 464 $\pm$ 76                | 522 $\pm$ 92                | 515 $\pm$ 67                | 489 $\pm$ 18                | 423 $\pm$ 19                |
> | Run      | 350 $\pm$ 69                | 444 $\pm$ 13                | 423 $\pm$ 53                | 345 $\pm$ 5                 | 266 $\pm$ 45                |
> | **Average** | 640                 | **693**                     | 686                     | 646                     | 551                     |
>
>
> **Ablation Study on $ \beta $.**
> Similarly, Table 6 reports the performance for different $ \beta $ values in the Walker domain. The results reveal that DVFB exhibits only minor deviations across different values. This demonstrates the robustness of our framework to changes in $ \beta $.
>
> **Table 6 Ablation Study on $ \beta $**
> | **Task** | $ \boldsymbol{\beta = 0.1} $ | $ \boldsymbol{\beta = 0.3} $ | $ \boldsymbol{\beta = 0.5} $ | $ \boldsymbol{\beta = 0.7} $ | $ \boldsymbol{\beta = 0.9} $ |
> |----------|-----------------------------|-----------------------------|-----------------------------|-----------------------------|-----------------------------|
> | Stand    | 819 $\pm$ 32                 | 862 $\pm$ 9                 | 905 $\pm$ 27                | 898 $\pm$ 62                | 919 $\pm$ 9                 |
> | Walk     | 819 $\pm$ 38                 | 861 $\pm$ 18                | 900 $\pm$ 53                | 926 $\pm$ 17                | 873 $\pm$ 32                |
> | Flip     | 428 $\pm$ 10                 | 501 $\pm$ 18                | 515 $\pm$ 67                | 616 $\pm$ 129               | 453 $\pm$ 30                |
> | Run      | 344 $\pm$ 28                 | 397 $\pm$ 40                | 423 $\pm$ 53                | 434 $\pm$ 54                | 342 $\pm$ 35                |
> | **Average** | 603                 | 655                     | 686                     | **719**                     | 647                     |
>
>
> **Ablation Study on $ \eta $.**
> We conduct an ablation study in the Quadruped domain to evaluate the sensitivity of the DVFB framework to the hyperparameter $ \eta $ during the fine-tuning phase. As shown in Table 7, the overall average performance across all tasks remains consistent, indicating that DVFB is relatively resilient to changes in $ \eta $.
>
> **Table 7 Ablation Study on $ \eta $**
> | **Task** | $ \eta = 0.02 $ | $ \eta = 0.1 $ | $ \eta = 0.5 $ | $ \eta = 1.0 $ |
> |----------|-----------------|----------------|----------------|----------------|
> | Stand    | 957 $\pm$ 4     | 964 $\pm$ 6    | 965 $\pm$ 7    | 954 $\pm$ 10   |
> | Walk     | 891 $\pm$ 32    | 908 $\pm$ 30   | 908 $\pm$ 21   | 886 $\pm$ 8    |
> | Jump     | 830 $\pm$ 18    | 838 $\pm$ 11   | 831 $\pm$ 20   | 835 $\pm$ 8    |
> | Run      | 557 $\pm$ 15    | 530 $\pm$ 15   | 536 $\pm$ 27   | 543 $\pm$ 26   |
> | **Average** | 809         | **810**         | 804         | 804         |
>
> The results of our sensitivity analysis demonstrate that DVFB is resilient to variations in key hyperparameters $ \alpha $, $ \beta $, and $ \eta $. Furthermore, for all other neural network hyperparameters (e.g., learning rate), we adopt the default settings of URL with DDPG, ensuring consistency with prior work.

---

> ### Author Response · Authors · 2024-11-22
> **Continued**
>
> ### **Q6. Could you elaborate on the possible limits of DVFB, including computational complexity and scalability in practical applications? Considering these criteria would yield a more equitable perspective on the approach's practical viability.**
>
> **A6.** Thank you for raising questions about DVFB's practical limitations. We conduct comprehensive experiments to analyze its computational complexity and scalability across different domains.
>
> ### **Computational Analysis.**
> We compare the training time of DVFB with several baseline methods on identical hardware (RTX 3090). As shown in Table 8, DVFB requires approximately 19 hours of pre-training time, which is moderate compared to other methods:
>
> **Table 8 Training time comparison across different methods**
> | **Method**  | **Training Time**  |
> |-------------|---------------------|
> | CIC         | ~ 12 hours          |
> | ComSD       | ~ 16 hours          |
> | BECL        | ~ 24 hours          |
> | CeSD        | ~ 29 hours          |
> | CL          | ~ 11 hours          |
> | LAP         | ~ 12 hours          |
> | LRA-P       | ~ 12 hours          |
> | LRA-SR      | ~ 12 hours          |
> | FB          | ~ 12 hours          |
> | **DVFB**    | ~ 19 hours          |
>
> While DVFB's training time is higher than some baselines due to its dual-value architecture, we believe this computational overhead is justified by its superior performance.
>
> ### **Further Domain Evaluation.**
> We evaluate DVFB's scalability across different domains, including point-mass maze and manipulation tasks. Table 9 shows the performance comparison. The results demonstrate DVFB's generalization capabilities across different environments, consistently outperforming baseline methods.
>
> **Table 9 Performance comparison across different domains**
> | **Domain**      | **Task**          | **FB**     | **CIC**    | **CeSD**   | **FB-offline*** | **MCFB-offline*** | **DVFB**   |
> |-----------------|-------------------|------------|------------|------------|-----------------|-------------------|------------|
> | **Point-Mass**  | Reach Top-left    | 69 ± 6     | 18 ± 6     | 12 ± 8     | 612             | 773               | 932 ± 10 |
> |                 | Reach Top-right   | 77 ± 95    | 5 ± 2      | 5 ± 4      | 0               | 270           | 203 ± 81   |
> |                 | Reach Bottom-left | 3 ± 3      | 7 ± 4      | 18 ± 21    | 268         | 1                 | 94 ± 45    |
> |                 | Reach Bottom-right| 0 ± 0      | 2 ± 2      | 2 ± 2      | 0               | 0                 | 4 ± 3  |
> |                 | **Average**       | 37.3   | 8.0    | 9.3    | 219             | 261               | **308.3**  |
> | **Meta-World**  | Faucet Open       | 0.18       | 0.04       | 0.00       | ---             | ---               | **0.60**   |
> |                 | Faucet Close      | 0.10       | 0.18       | 0.00       | ---             | ---               | **0.52**   |
>
> *Results marked with * are sourced from MCFB (NeurIPS 24) [3].*
>
> ### **Practical Limitations.**
> In summary, although DVFB demonstrates exceptional zero-shot performance in simulation environments, it still has room for further optimization in terms of computation time. In addition, the effectiveness of DVFB in varied real-world contexts such as quadruped robot control, robot manipulation, and so on is still unknown and needs to be further explored.
>
> ---
>
> ### References
>
> [1] Yang Y, Zhou T, He Q, et al. Task Adaptation from Skills: Information Geometry, Disentanglement, and New Objectives for Unsupervised Reinforcement Learning. ICLR, 2024.
>
> [2] Touati A, Rapin J, Ollivier Y. Does zero-shot reinforcement learning exist?. ICLR, 2023.
>
> [3] Jeen S, Bewley T, Cullen J. Zero-Shot Reinforcement Learning from Low Quality Data. NeurIPS, 2024.

---

> > ### Comment · Reviewer_pWXy · 2024-11-25
> >
> > Thank you for your thorough revisions. I appreciate that you have addressed most of the points I raised in my review. As a result, I have increased my overall rating from 5 to 6. Additionally, I have raised the presentation score from 2 to 3 due to the improvements in the clarity of the figures and pseudocode. Finally, I hope you will consider sharing your code with the research community, as it would greatly benefit others working in this area. Best regards.

---

> > > ### Author Response · Authors · 2024-11-26
> > > **Thank you!**
> > >
> > > Thank you for taking the time to help us improve our work. We are pleased to see that we could address your concerns. We sincerely appreciate you raising the score on our work. We are currently finalizing the code and its documentation to ensure ease of use and plan to release it publicly upon the paper’s acceptance. Once again, thank you for your constructive feedback and support!

---

> ### Author Response · Authors · 2024-11-24
> **Looking forward to further discussions!**
>
> Dear Reviewer pWXy,
>
> Thank you for your insightful comments. We were wondering if our response and revision have resolved your concerns. We have attempted to address your initial questions through our replies and are eager to clarify any further points you might raise. Please feel free to provide additional feedback. We greatly appreciate your continued engagement.
>
> Best regards, Authors

---

### Official Review · Reviewer_yJG4 · 2024-11-04

**Soundness:** 3
**Presentation:** 4
**Contribution:** 3
**Rating:** 8
**Confidence:** 3

**Summary:**

This work introduces the Dual-Value Forward-Backward (DVFB) representation framework for unsupervised reinforcement learning (URL). It tackles the challenge of enabling agents to generalise to new tasks without further training (zero-shot generalisation) in addition to fine-tuning adaptation in online settings.

It builds on successor representation (SR)-based approaches which aim to learn a representation of expected future states and have been shown to aid zero-shot generalisation in RL. In particular, the work extends forward-backward (FB) representations by learning both forward and backward dynamics. The authors explore failures in FB-based approaches in online URL settings and find that it is due to inadequate exploration. They address this by introducing an intrinsic reward based on contrastive learning to encourage exploration, combining this “exploration value” function to the usual “skill value” function to arrive at their DVFB. The authors also introduce a fine-tuning scheme using a reward mapping technique to add further online adaptation capabilities to their method.

The authors validate DVFB across twelve robot control tasks in the DeepMind Control Suite and demonstrate the approach gives both superior zero-shot and fine-tuning performance.

**Strengths:**

- Zero-shot generalisation in online settings is an important problem in RL, where progress is going to be essential for successfully deploying RL in real-world applications. DVFB advances the field's understanding of how to create agents that can solve and adapt to new tasks immediately, without requiring extensive retraining.
 - The authors build on foundational concepts in the field such as SR and FB representations. The paper does a good job identifying the limitations of FB in online settings, pinpointing insufficient exploration as a core issue, and using their insights to justify the extensions of FB into DVFB. The introduction of a novel exploration value function alongside the skill value function is an original approach that enhances exploration and, as shown in their motivation and results, improves generalisation. Furthermore, the addition of a reward mapping is a valuable addition that enables them to demonstrate both zero-shot generalisation and fine-tuning in an online setting.
 - Impressive results: the paper presents rigorous experiments across 12 diverse control tasks in a widely used benchmark for tasks requiring fine-grained motor control. In terms of zero-shot performance, their method outperforms baseline methods across the 12 tasks, particularly in tasks where others struggle with exploration (further supporting their motivation). It also outperforms on fine-tuning performance, showing faster adaptation and greater stability compared to the state-of-the art in URL.
 - The paper is well written, guiding the reader through the problem being addressed, relevant related work, the motivations for their extensions, implementation, results and conclusions. The methodology section is nicely laid out, with clear explanations and schematics detailing components of the model. Their experimental results are clearly presented and easy to understand.

**Weaknesses:**

- Potential for broader applicability: the paper focuses on tasks in the DeepMind Control Suite. This demonstrates DVFB’s capability in robotic control tasks, but leaves one wondering about the framework's versatility which otherwise seems very general. Could the authors discuss the potential for adapting DVFB to other domains, such as navigation? If possible, preliminary results or  discussion on expected performance in different contexts would broaden the scope of the work.
 - Other intrinsic rewards: the paper attributes improvements to enhanced exploration, but it doesn’t delve into specific advantages that contrastive entropy is providing over other intrinsic rewards. Going beyond the DVFV w/o CE ablation experiment and trying out other intrinsic rewards (beyond just RND rewards) could add further insight into their particular choice of contrastive entropy.
 - Sparse presentation of reward mapping technique: there’s limited detail and justification for the reward mapping technique. It’s unclear whether this particular mapping method is optimal or if other strategies might perform equally well or even better in different tasks.  Further exploration would clarify its effectiveness and limitations. Could the authors discuss more justification for this approach, as well as analysing some alternatives?
 - Reproducibility: lack of code to reproduce the results: providing code would significantly enhance the paper’s accessibility. While the inclusion of pseudocode and hyperparameters is appreciated and provides important details for the method, the absence of actual code makes it challenging for others to fully replicate the experiments or apply the DVFB framework to other settings.

**Questions:**

Reflecting the weaknesses discussed above, my key questions are:
 - How broadly applicable is the method, particularly beyond robotic control tasks? Are there any preliminary results in other domains that the authors could include?
 - How important is their particular choice of reward to encourage exploration -- the contrastive entropy reward? How well would other rewards stand-in for this, or is it particularly well suited?
 - Similar questions for the reward mapping technique. Could we see more justification for their approach and other alternatives explored?
 - Can the authors provide code so that others can directly reproduce the results?

---

> ### Author Response · Authors · 2024-11-22
> **Author response to review by reviewer yJG4**
>
> Thank you for your thoughtful and constructive feedback, which will greatly help in refining our work and expanding its applicability. Here are our detailed responses.
>
> ### **Q1. How broadly applicable is the method, particularly beyond robotic control tasks? Are there any preliminary results in other domains that the authors could include?**
>
> **A1.** Thank you for your insightful feedback regarding the broader applicability of DVFB beyond the DeepMind Control Suite. According to your suggestion, we conduct additional experiments on Point-Mass Maze navigation environment and Meta-World robotic manipulation environment. Although the offline methods FB-offline[1] and MCFB-offline[2] rely on pre-collected offline datasets (in contrast to online URL, high data sensitivity of offline zero-shot methods introduces a significant challenge in collecting suitable offline datasets), we provide a comparison with these offline settings for a more comprehensive evaluation of our approach. Since the Meta-World experiment is not performed in these methods, we summarize the results of FB-offline and MCFB-offline with RND offline data in Point-Mass Maze domain. As shown in Table 1, DVFB outperforms baseline methods across both domains. The additional results demonstrate that DVFB is not only effective in robotic control tasks but also generalizes well to other domains, such as navigation and robotic manipulation.
>
>
> **Table 1: Performance comparison across different domains. For Point-Mass Maze, results show mean ± standard deviation across three seeds. For Meta-World, results show success rates.**
> | Domain        | Task             | FB      | CIC     | CeSD    | FB-offline* | MCFB-offline* | DVFB       |
> |---------------|------------------|---------|---------|---------|-------------|---------------|------------|
> | **Point-Mass**| Reach Top-left    | 69 ± 6  | 18 ± 6  | 12 ± 8  | 612         | 773           | 932 ± 10|
> |               | Reach Top-right   | 77 ± 95 | 5 ± 2   | 5 ± 4   | 0           | 270       | 203 ± 81   |
> |               | Reach Bottom-left | 3 ± 3   | 7 ± 4   | 18 ± 21 | 268     | 1             | 94 ± 45    |
> |               | Reach Bottom-right| 0 ± 0   | 2 ± 2   | 2 ± 2   | 0           | 0             | 4 ± 3  |
> |               | **Average**       | 37.3    | 8.0     | 9.3     | 219         | 261           | **308.3**  |
> | **Meta-World**| Faucet Open       | 0.18    | 0.04    | 0.00    | ---         | ---           | **0.60**   |
> |               | Faucet Close      | 0.10    | 0.18    | 0.00    | ---         | ---           | **0.52**   |
>
> *Results marked with * are sourced from MCFB (NeurIPS 24).*

---

> ### Author Response · Authors · 2024-11-22
> **Continued**
>
> ### **Q2. How important is their particular choice of reward to encourage exploration -- the contrastive entropy reward? How well would other rewards stand-in for this, or is it particularly well suited?**
>
> **A2.** Thank you for this valuable suggestion about comparing different intrinsic rewards. Our contrastive entropy reward is designed to encourage skill discrimination during exploration, which keeps skills learned by the FB mechanism. To evaluate the role of the contrastive entropy reward, we conduct two sets of comprehensive experiments.
>
> **Ablation study.** We perform an ablation study on the coefficient $\beta$ of contrastive entropy reward $r_{intr}=r_{rnd}+\beta r_{ce}$, as shown in Table 2. The results demonstrate that increasing $\beta$ from 0.1 to 0.7 consistently improves performance, validating that contrastive entropy enhances generalization by promoting skill separability. However, further increasing $\beta$ negatively impacts performance due to an imbalance between skill learning and exploration.
>
> **Table 2: Ablation study on contrastive entropy coefficient $\beta$ on Walker tasks. Results show mean ± standard deviation across three seeds.**
> | Task   | $\beta$=0.1 | $\beta$=0.3 | $\beta$=0.5 | $\beta$=0.7 | $\beta$=0.9 |
> |--------|-------------|-------------|-------------|-------------|-------------|
> | Stand  | 819±32      | 862±9       | 905         | 898±62      | 919±9       |
> | Walk   | 819±38      | 861±18      | 900         | 926±17      | 873±32      |
> | Flip   | 428±10      | 501±18      | 515         | 616±129     | 453±30      |
> | Run    | 344±28      | 397±40      | 423         | 434±54      | 342±35      |
> | **Average** | 603       | 655         | 686         | **719**     | 647         |
>
> **Comparison experiment.** Following your suggestion, we compare DVFB with variants using alternative intrinsic rewards (ICM-APT, Proto, and CIC), as shown in Table 3.
>
> **Table 3: Comparison with alternative intrinsic rewards on Walker tasks. Results show mean ± standard deviation across three seeds.**
>
> | Task   | DVFB(ICM-APT) | DVFB(Proto) | DVFB(CIC) | DVFB      |
> |--------|---------------|-------------|-----------|-----------|
> | Stand  | 883±106       | 844±101     | 846±74    | **905±27**|
> | Walk   | 840±85        | 821±27      | 825±24    | **900±53**|
> | Flip   | 436±68        | 454±51      | 436±140   | **515±67**|
> | Run    | 354±15        | 358±17      | 342±14    | **423±53**|
> | **Average** | 628       | 619         | 612       | **686**   |
>
> The results reveal several key insights:
> 1. **The scalability of DVFB.** All variants achieve reasonable zero-shot generalization performance, demonstrating DVFB's compatibility with different intrinsic rewards.
> 2. **The advantage of CE reward.** DVFB with contrastive entropy consistently outperforms other variants, achieving the highest average performance.
>
> These experiments provide strong evidence for the effectiveness of the proposed contrastive entropy reward, while also demonstrating DVFB's flexibility in incorporating various intrinsic rewards.

---

> ### Author Response · Authors · 2024-11-22
> **Continued**
>
> ### **Q3. Similar questions for the reward mapping technique. Could we see more justification for their approach and other alternatives explored?**
>
> We appreciate the reviewer’s thoughtful feedback on the reward mapping technique. In response, we provide a detailed justification and explore potential alternatives during our experiments.
>
> **Potential fine-tuning techniques.** For fine-tuning, our objective is to ensure stable policy improvements over zero-shot performance. The reward mapping is designed to achieve stable improvement by balancing the importance of prior skill value $Q_M$ and downstream task value $Q_F$. A straightforward alternative is directly using the downstream task rewards $r_t$ for the task value, and choosing a suitable coefficient $\eta$ in Eq.11 to balance the value of $Q_M$ and $Q_F$. We call it DVFB w/o MAP, using dual-value fine-tuning with the downstream task value based on downstream task rewards $r_t$ rather than $r_f$. We also explore an adaptive method to dynamically adjust the $\eta$ parameter by setting $\eta = \frac{Q}{MQ}$, which we refer to as DVFB w/o MAP adaptive.
>
> **Experimental results.** We conduct comparative experiments in the quadruped domain, with the results presented in https://sites.google.com/view/figureeta and Table 4. The findings show that DVFB w/o MAP achieves stable but limited improvement with higher coefficients, while exhibiting unstable performance with lower coefficients, ultimately constraining overall performance. It is also challenging for DVFB w/o MAP adaptive to achieve superior improvement due to the nonlinear relationship between skill value and downstream value. In contrast, DVFB with the reward mapping technique consistently delivers stable and superior improvements across all tasks. These results validate the effectiveness of our chosen approach.
>
> **Table 4: Performance comparison of different fine-tuning approaches on the quadruped domain.
> Results show mean ± std over three runs.**
>
> | Task   | $\eta = 0.02$ | $\eta = 0.1$ | $\eta = 0.5$ | adaptive $\eta$ | DVFB     |
> |--------|----------------|--------------|--------------|-----------------|----------|
> | Stand  | 954$\pm$5      | 951$\pm$10   | 961$\pm$8    | 960$\pm$5       | 965$\pm$7|
> | Walk   | 753$\pm$32     | 765$\pm$18   | 752$\pm$31   | 820$\pm$77      | 908$\pm$21|
> | Jump   | 819$\pm$28     | 811$\pm$16   | 784$\pm$59   | 830$\pm$12      | 831$\pm$20|
> | Run    | 496$\pm$23     | 491$\pm$4    | 490$\pm$14   | 496$\pm$48      | 536$\pm$27|
> | **Average** | 756         | 755          | 747          | 777             | **804**      |
>
> **Justification for the reward mapping technique.**
> The reward mapping mechanism in DVFB helps balance prior skill value and downstream task value, enabling stable and effective fine-tuning. Our experiments demonstrate that this approach significantly outperforms alternatives. The consistent improvement across tasks provides strong evidence for the effectiveness of this technique. We agree that exploring other effective alternative options is a meaningful topic.
>
> ---
>
> ### **Q4. Can the authors provide code so that others can directly reproduce the results?**
>
> **A4.** Thank you for your suggestion regarding code availability to ensure reproducibility. We are committed to making all relevant code and implementation details publicly available upon acceptance of the paper to facilitate reproducibility and further research.  We appreciate your understanding and support for this approach.
>
> ---
>
> ### References
>
> 1. Touati A, Rapin J, Ollivier Y. Does zero-shot reinforcement learning exist?. *ICLR*, 2023.
> 2. Jeen S, Bewley T, Cullen J. Zero-Shot Reinforcement Learning from Low Quality Data. *NeurIPS*, 2024.

---

> ### Author Response · Authors · 2024-11-24
> **Looking forward to further discussions!**
>
> Dear Reviewer yJG4,
>
> Thank you for your insightful comments. We were wondering if our response and revision have resolved your concerns. We have attempted to address your initial questions through our replies and are eager to clarify any further points you might raise. Please feel free to provide additional feedback. We greatly appreciate your continued engagement.
>
> Best regards, Authors

---

> > ### Comment · Reviewer_yJG4 · 2024-11-25
> >
> > Many thanks for the detailed responses to my and the other reviewers comments and concerns. The extra experiments and ablations are interesting and fill in the gaps in presentation. It's also very promising to see that DVFB is effective in tasks beyond robotic control. Given the extra results and proposed changes, I am confident with my original scoring and still believe that this a good paper that would be a valuable addition to ICLR.

---

> > > ### Author Response · Authors · 2024-11-26
> > > **Thank you!**
> > >
> > > Thank you for your thoughtful feedback and for taking the time to assess our revisions. We are glad to see that we could address your concerns. We sincerely appreciate your recognition of the paper’s contribution and your positive evaluation of our work!

---

### Official Review · Reviewer_U97w · 2024-11-04

**Soundness:** 3
**Presentation:** 3
**Contribution:** 2
**Rating:** 6
**Confidence:** 4

**Summary:**

This work presented a pre-training framework for zero-shot reinforcement learning by leveraging forward-backward (FB) representation. Unlike some previous study on zero-shot RL, this work analysed the performance gap of FB in the online learning setting compared with a specific exploration strategy. The authors then proposed a new exploration reward based on contrastive learning and incorporated this into FB traning by end-to-end online learning. The proposed method is evaluated in zero-shot online URL and fine tuning settings. Experimental results suggest that it achieved improved performance than some baseline methods given limited interactions.

**Strengths:**

This work is well motivated and the promotion of exploratory behaviour during the pre-training phase to increase the data coverage is reasonable.

The paper is well written and easy to follow.

**Weaknesses:**

The major contribution in this work is the combination of an exploration reward with FB learning, where the technical novelty is limited.

Although the performance gain shown in Table 1 looks strong, I have concern on the baselines used in comparison for this setting. It is unclear why these are suitable baselines here for the problem of zero-shot online URL. Many baselines here are either not designed for online learning with self-generated trajectory (for example, LRA-SR used in (Touati et al., 2023)) or not zero-shot testing (if I’m not mistaken some baseline finetunes longer steps, for example, CeSD with 100k interactions rather than 1e^4 used in this work). So it does not look so exciting when explicit exploration techniques are used in combination with a zero-shot technique. A naive approach for this problem would be using a method of pure exploration (e.g. r_{ce} proposed in this work as the proposed intrinsic reward itself has the capability to collect an offline dataset.) or a method of skill discovery to collect an offline dataset with better data coverage than FB, then training FB on top of this dataset and testing its ability in zero-shot generalisation. This could probably better demonstrate the advantage of combining exploration reward with FB in online URL.

Following the previous comment, for Table 1, it would be better to group the baselines into several categories so that it is clear from the table which property (zero-shot, online, offline,  exploration or skill discovery) each method has or does not have.

There is no theoretical analysis to support the proposed objective function and reward function either in the FB pre-training stage or fine-tuning stage. It is unclear what guarantee of zero-shot generalisation of the proposed method can have.

Questions and suggestions:

For Fig. 2 and Fig. 3, Please add more descriptions to the caption so that the reader can understand the main discovery and meaning of the figure.

**Questions:**

Please see the weaknesses part.

---

> ### Author Response · Authors · 2024-11-22
> **Author response to review by reviewer U97w**
>
> Thank you very much for your insightful and constructive feedback on our submission. The following are our detailed responses.
>
> ### **Q1. The major contribution in this work is the combination of an exploration reward with FB learning, where the technical novelty is limited.**
>
> **A1.** As the reviewer pointed out, our method combines exploration rewards with FB learning. However, we would like to emphasize that this combination presents non-trivial technical challenges. Specifically, the value functions in FB [1] are inherently tied to implicit rewards for skill learning, and directly incorporating online exploration rewards could disrupt this skill learning capability. Our significant contributions include:
>
> **Scientific finding**: We systematically demonstrate that insufficient exploration in FB leads to low data diversity, which in turn results in inaccurate successor measures and limits zero-shot generalization. Our key scientific finding is that the critical challenge of online FB lies in preserving efficient FB representation learning while simultaneously increasing online data diversity.
>
> **Technical novelty**: To address this, we develop a novel DVFB framework that enhances exploration while maintaining effective skill learning through dual value functions and a novel intrinsic reward. Additionally, we introduce a dual-value fine-tuning scheme that achieves stable performance improvements, outperforming current SOTA methods.
>
> To the best of our knowledge, **DVFB is the first approach to successfully enable both zero-shot generalization and fine-tuning capabilities in online URL settings.** We believe our insights into extending successor representation methods for online URL provide solid contributions to the field. We hope that our novelty and distinct contributions have been adequately justified.
>
> ---
>
> ### **Q2. Although the performance gain shown in Table 1 looks strong, I have concern on the baselines used in comparison for this setting. It is unclear why these are suitable baselines here for the problem of zero-shot online URL.**
>
> **A2.** Thanks for raising this important concern. To the best of our knowledge, no existing method achieves zero-shot generalization in online URL. We selected the baselines because they represent the closest existing approaches to our problem setting: **zero-shot generalization in online URL**.
>
> **USD methods** (e.g., BeCL, CeSD) are designed for online URL and aim to learn transferable skills. While zero-shot generalization would be ideal, they typically rely on fine-tuning in downstream tasks due to limitations in mutual information-based skill learning [2]. Our results in Table 1 demonstrate that DVFB achieves zero-shot performance comparable to their fine-tuned performance (for more results, please refer to Table 2 in the attached PDF) and outperforms them in fine-tuning.
>
>  **Table 1: Comparison of Zero-shot and Fine-tuned Performance Across Domains**
>
> | Domain | Zero shot | BeCL | CeSD | DVFB | Fine-tune | BeCL* | CeSD* | DVFB |
> |--------|-----------|------|------|------|-----------|-------|-------|------|
> | Walker (average) |     | 59 | 113 | **686** |   | 708 | 668 | **852** |
> | Quadruped (average) |    | 103 | 375 | **715** |    | 720 | 787 | **804** |
> | Hopper (average) |     | 1 | 3 | **101** |   | 20 | 62 | **163** |
>
>
> *Results marked with **\*** are sourced from BeCL (ICML 23) and CeSD (ICML 24).*
>
> **SR methods** (e.g., LRA-SR, FB), originally designed for offline settings, represent the SOTA in offline zero-shot generalization. To ensure a fair comparison of zero-shot capabilities in online URL, we implemented their online versions based on the authors' official code.
>
> Our experimental setup ensures a fair comparison by evaluating all methods under identical conditions (same interaction steps and fine-tuning steps), demonstrating DVFB's superior performance in both zero-shot and fine-tuning scenarios.

---

> ### Author Response · Authors · 2024-11-22
> **Continued**
>
> ### **Q3. A naive approach for this problem would be using a method of pure exploration or a method of skill discovery to collect an offline dataset with better data coverage than FB, then training FB on top of this dataset and testing its ability in zero-shot generalisation.**
>
> **A3.** We appreciate your comment for considering zero-shot offline baselines (i.e., a method of skill discovery to collect an offline dataset, then training FB on top of this dataset). Following your suggestion, we evaluate DVFB against SR methods, including LAP, LRA-SR, LRA-P, and FB, across the Walker, Quadruped, and Cheetah domains under offline settings. The performances of offline methods are from FB [1], where offline datasets are collected by APS, Proto, and RND exploration methods. The results, summarized in  Appendix G (also see https://sites.google.com/view/zero-shot-generalization-perfo), highlight the following key differences between offline and online methods:
>
> **High data sensitivity for offline methods:** Offline methods exhibit significant performance variation depending on the quality of exploration data. On the one hand, *the same algorithm requires different exploration datasets in different domains.*
> For example, FB trained with Proto data in the Walker domain achieves best performance (666), while in the Quadruped domain using Proto data yields a huge performance drop (222). On the other hand, *different algorithms require different exploration datasets.* For example, LRA-P performs best with RND data, while FB performs best with APS data. **_When designing a novel algorithm, how to make sure what kind of exploration dataset is most suitable?**
> An intuitive idea is to train the models on different exploration datasets, and compare to find the best performance models. Obviously, it will lead to **high computational costs**.
> In contrast, DVFB does not depend on offline datasets, and requires only a single agent pre-training phase to achieve strong zero-shot capability, significantly reducing time and computational overhead. It is simpler and easier to deploy than offline zero-shot methods.
>
> **Performance limitation on pre-collected fixed dataset:** Offline methods rely on the diversity and quality of fixed pre-collected datasets, which limits their generalization performance. In contrast, DVFB balances the exploration and exploitation online with an intrinsic reward based on contrastive learning, leading to enhanced skill learning and better zero-shot performance. Experimental results across **twelve tasks in Mujoco domains** demonstrate that DVFB consistently outperforms zero-shot offline methods.
>
> In summary, the results demonstrate that DVFB offers superior performance and efficiency compared to both offline and online methods, establishing its significance in zero-shot online URL.
>
> ---
> ### **Q4. Following the previous comment, for Table 1, it would be better to group the baselines into several categories so that it is clear from the table which property (zero-shot, online, offline, exploration or skill discovery) each method has or does not have.**
>
> **A4.** We agree that grouping the baselines according to their properties will enhance clarity. We categorize the methods into several groups based on their key properties as shown in the table below. We hope this categorization makes it easier to understand the key properties of each method and their relationships.
>
> **Table 2 Properties of Different Methods**
>
> | Method               | Publish    | Zero-shot | Online | Offline | Exploration | Skill Discovery |
> |----------------------|------------|-----------|--------|---------|-------------|-----------------|
> | **Successor Representation Methods** |            |           |        |         |             |                 |
> | CL                   | NeurIPS 22 | ✔         | ✘      | ✔       | ✘           | ✔               |
> | Lap                  | ICLR 18   | ✔         | ✘      | ✔       | ✘           | ✔               |
> | LRA-P                | ICLR 23   | ✔         | ✘      | ✔       | ✘           | ✔               |
> | LRA-SR               | ICLR 23   | ✔         | ✘      | ✔       | ✘           | ✔               |
> | FB                   | ICLR 23   | ✔         | ✘      | ✔       | ✘           | ✔               |
> | **Unsupervised Skill Learning Methods** |            |           |        |         |             |                 |
> | CIC                  | NeurIPS 22 | ✘         | ✔      | ✘       | ✔           | ✔               |
> | BeCL                 | ICML 23   | ✘         | ✔      | ✘       | ✔           | ✔               |
> | ComSD                | Arxiv 23  | ✘         | ✔      | ✘       | ✔           | ✔               |
> | CeSD                 | ICML 24   | ✘         | ✔      | ✘       | ✔           | ✔               |
> | **DVFB (ours)**      |            | ✔         | ✔      | ✘       | ✔           | ✔               |

---

> ### Author Response · Authors · 2024-11-22
> **Continued**
>
> ### **Q5. It is unclear what guarantee of zero-shot generalization of the proposed method can have.**
>
> **A5.** Thank you for your insightful feedback. We agree that providing a theoretical foundation for the proposed method is crucial to support its objectives and claims. Below, we address your concerns by offering a theoretical guarantee for zero-shot generalization and a detailed analysis of how the DVFB framework improves FB’s zero-shot generalization in online URL.
> Theoretical guarantee for zero-shot generalization and theoretical analysis is shown in  Appendix H (also see https://sites.google.com/view/theoretical-gurantee).
>
> According to our experimental results, the DVFB framework consistently outperforms FB across diverse unseen tasks. These findings confirm the zero-shot generalization capability of the DVFB framework in online URL.
>
> ---
>
> ### **Q6. For Fig. 2 and Fig. 3, Please add more descriptions to the caption so that the reader can understand the main discovery and meaning of the figure.**
>
> **A6.** Thank you for your suggestion. We agree that the captions for Figures 2 and 3 could be clearer. We revise them as follows:
>
> **Figure 2**: The $x$ axis "position" means the walking distances for different trajectories, and short-distance trajectories reflect less diverse exploration outcomes. The $y$ axis "skill index" means the index of different skills. Lines of the same color represent trajectories corresponding to the same skill. The revised caption is:
>
> > "The trajectories of the RND, FB-online, and FB-offline agents in the *Walker walk* task. The FB-online agent learns short-range trajectories, in contrast to the RND agent's diverse exploration and the FB-offline agent's ability to master long-range locomotion skills."
>
> **Figure 3**: In (a), the x-axis represents time steps during pre-training, and the y-axis shows the episode rewards of agents. In (b), the x-axis represents trajectories categorized by episode reward ranges. The y-axis in the top figure indicates the skill values, while the y-axis in the bottom figure depicts the Spearman correlation between skill values and episode rewards. The revised caption is:
>
> > "Generalization curve and value function properties. (a) presents the agents' performance during pre-training, while (b) illustrates the normalized skill value and the Spearman correlation between skill value and return across different return ranges."
>
> We have updated the captions and explanations of figures in the revised PDF.
>
>
> ---
>
> ### References
>
> 1. Touati A, Rapin J, Ollivier Y. Does zero-shot reinforcement learning exist?. ICLR, 2023.
>
> 2. Yang Y, Zhou T, He Q, et al. Task Adaptation from Skills: Information Geometry, Disentanglement, and New Objectives for Unsupervised Reinforcement Learning. ICLR, 2024.

---

> ### Author Response · Authors · 2024-11-24
> **Looking forward to further discussions!**
>
> Dear Reviewer U97w,
>
> Thank you for your insightful comments. We were wondering if our response and revision have resolved your concerns. We have attempted to address your initial questions through our replies and are eager to clarify any further points you might raise. Please feel free to provide additional feedback. We greatly appreciate your continued engagement.
>
> Best regards, Authors

---

> ### Author Response · Authors · 2024-11-26
> **Looking forward to further discussions!**
>
> Dear Reviewer U97w,
>
> Thank you for taking the time to review our manuscript and for providing valuable suggestions.  We have further revised the main PDF to address your main concerns more clearly, with the changes highlighted in blue.
>
> Below are the specific changes we made in response to your feedback:
> 1. We explain the contributions of DVFB and the scientific findings related to extending FB to online URL.
> 2. We discuss the limitations of the offline zero-shot method in Section 2 (RELATED WORK) and highlight DVFB’s advantages over SOTA offline zero-shot methods in Appendix G.
> 3. We provide a clear categorical description of the baselines in Appendix C.2 and offer guidance in Section 6 (EXPERIMENTS).
> 4. We present a detailed theoretical analysis of how the DVFB framework improves online FB’s zero-shot generalization in Appendix H, and provide guidance in Section 5 (METHODOLOGY).
> 5. We have updated the captions and explanations of the figures in Section 4.
> 6. We conduct additional experiments on the Point-Mass Maze and Meta-World benchmarks in Appendix G, along with detailed ablations in Appendix K (reward mapping technique) and Appendix L (sensitivity to hyperparameters) to further demonstrate the effectiveness of DVFB.
>
> As the rebuttal period is closing, the authors would greatly appreciate it if the reviewer could consider our responses to the original review. We are more than delighted to have further discussions and improve our manuscript. If our responses have addressed your concerns, we would be grateful if you could kindly re-evaluate our work.
>
> Best regards, Authors

---

> > ### Comment · Reviewer_U97w · 2024-11-27
> >
> > I would like to thank the authors for providing further clarifications and additional experimental results. These results have addressed most of my concerns. The paper has also shown improvements after the revision. Consequently, I have increased my overall score by 1.

---

> > > ### Author Response · Authors · 2024-11-27
> > > **Thank you!**
> > >
> > > Thank you for taking the time to review our revised manuscript and for your valuable feedback throughout the review process. We sincerely appreciate your recognition of the improvements made and are glad to hear that the additional clarifications and experimental results have addressed your concerns. Your constructive comments have been instrumental in enhancing the quality of our work, and we are grateful for your support and encouragement.

---

### Author Response · Authors · 2024-11-22
**Meta response from the authors**

We sincerely thank all reviewers for their constructive feedback (we refer to U97w as R1, yJG4 as R2, and pWXy as R3). We are grateful that **most reviewers positively acknowledge our overall contributions**, such as *"identifying the limitations of FB in online settings"* and *"a novel exploration value... is an original approach"* (R2), as well as *"presents a novel DVFB paradigm... providing an innovative method for..."* (R3). Furthermore, the **superior performance** of our approach has been highlighted: *"achieved improved performance"* (R1) and *"impressive results, faster adaptation and greater stability"* (R2).

To address the concerns raised, we make a series of major revisions, summarized below:

1. **Appendix C.2**: We categorize the baseline methods based on their properties, facilitating a clearer understanding of their key characteristics and interrelationships.

2. **Appendix G**: We conduct new experiments in the DMC benchmark to compare with popular offline zero-shot methods and analyze the advantages of zero-shot online URL over zero-shot offline URL.

3. **Appendix H**: We present a theoretical analysis to substantiate the zero-shot generalization capability of our proposed method.

4. **Appendix I**: We extend the evaluation of DVFB to additional navigation and robotic manipulation domains, showcasing its potential for broader applicability.

5. **Appendix J**: We conduct a comparative study of the contrastive entropy reward against other reward functions, including ICM-APT, Proto, and CIC, to analyze its significance and impact.

6. **Appendix K**: We provide a detailed implementation of the reward mapping technique and analyze its significance through comparisons with alternative approaches.

7. **Appendix L**: We perform additional ablation studies to evaluate the sensitivity of DVFB to hyperparameters.

8. **Figures and Explanations**: We revise the figures in the paper and add detailed explanations for both the figures and the experiments to improve the readers' understanding.

In summary, we have **significantly extended the empirical evaluations** in the revised manuscript. Importantly, the results of these comprehensive experiments are **generally consistent with the observations and conclusions from our original submission**. Please refer to the revised manuscript (highlighted changes are in blue) for details. Additionally, the responses to specific reviewer comments offer further clarification on these updates.

Please let us know if we have sufficiently addressed your concerns. We are happy to engage in further discussions and make additional improvements. If our response meets your expectations, we would greatly appreciate a re-evaluation of our work.

---

### Meta-Review · Area_Chair_ct8H · 2024-12-15

**Metareview:**

This paper introduces the Dual-Value Forward-Backward (DVFB) framework for unsupervised reinforcement learning (URL), aimed at achieving both zero-shot generalization and fine-tuning adaptation. The method combines a skill value function with an exploration value function to improve data diversity and generalization, addressing the limitations of forward-backward representations in online settings. A contrastive entropy intrinsic reward enhances exploration, while a dual-value fine-tuning scheme optimizes performance on downstream tasks. Experimental results show that DVFB outperforms existing methods in both zero-shot generalization and fine-tuning across multiple tasks.

The strengths of this paper include a well-motivated and clear method, impressive results (including additional results provided during the rebuttal), and clear writing. The main weaknesses of this paper include concerns about the novelty of combining exploration reward with FB learning, as raised by U97w, and justifications for the reward and reward mapping techniques.

The three reviews are all positive (6, 6, and 8), suggesting acceptance of this paper.

**Additional Comments On Reviewer Discussion:**

Reviewer U97w asked for more evaluation results, and the authors provided additional results, which led the reviewer to increase their score.
Reviewer pWXy also raised their rating due to the extra results and improved presentation.
Both reviewers yJG4 and pWXy requested that the code be released, and the authors have promised to do so.

---

### Decision · Program_Chairs · 2025-01-22

Accept (Poster)